# Regulation of BMP4/Dpp retrotranslocation and signaling by deglycosylation

Antonio Galeone[1,2], Joshua M Adams[3,4], Shinya Matsuda[5], Maximiliano F Presa[6], Ashutosh Pandey[1], Seung Yeop Han[1], Yuriko Tachida[7,8], Hiroto Hirayama[7,8], Thomas Vaccari[2], Tadashi Suzuki[7,8], Cathleen M Lutz[6], Markus Affolter[5], Aamir Zuberi[6], Hamed Jafar-Nejad[1,3]*

[1]Department of Molecular and Human Genetics, Baylor College of Medicine, Houston, United States; [2]Department of Biosciences, University of Milan, Milan, Italy; [3]Program in Developmental Biology, Baylor College of Medicine, Houston, United States; [4]Medical Scientist Training Program, Baylor College of Medicine, Houston, United States; [5]Biozentrum, University of Basel, Basel, Switzerland; [6]The Jackson Laboratory, Bar Harbor, United States; [7]Glycometabolome Biochemistry Laboratory, RIKEN Cluster for Pioneering Research, Saitama, Japan; [8]T-CiRA joint program, Kanagawa, Japan

*For correspondence:
hamedj@bcm.edu

Competing interests: The authors declare that no competing interests exist.

**Abstract** During endoplasmic reticulum-associated degradation (ERAD), the cytoplasmic enzyme *N*-glycanase 1 (NGLY1) is proposed to remove *N*-glycans from misfolded *N*-glycoproteins after their retrotranslocation from the ER to the cytosol. We previously reported that NGLY1 regulates *Drosophila* BMP signaling in a tissue-specific manner (Galeone et al., 2017). Here, we establish the *Drosophila* Dpp and its mouse ortholog BMP4 as biologically relevant targets of NGLY1 and find, unexpectedly, that NGLY1-mediated deglycosylation of misfolded BMP4 is required for its retrotranslocation. Accumulation of misfolded BMP4 in the ER results in ER stress and prompts the ER recruitment of NGLY1. The ER-associated NGLY1 then deglycosylates misfolded BMP4 molecules to promote their retrotranslocation and proteasomal degradation, thereby allowing properly-folded BMP4 molecules to proceed through the secretory pathway and activate signaling in other cells. Our study redefines the role of NGLY1 during ERAD and suggests that impaired BMP4 signaling might underlie some of the NGLY1 deficiency patient phenotypes.

## Introduction

*N*-Glycanase 1 (NGLY1; also known as peptide:*N*-glycanase or PNGase) is a cytoplasmic enzyme capable of removing *N*-glycans from glycoproteins (*Suzuki et al., 2002*). Mutations in human *NGLY1* cause an autosomal recessive, multi-system developmental disorder called NGLY1 deficiency (OMIM # 610661) (*Need et al., 2012*; *Enns et al., 2014*). NGLY1 and its homologs recognize and cleave *N*-glycans from their target proteins, changing the asparagine (N) to aspartic acid (D) upon removing the sugar chain (*Hirayama et al., 2015*). *N*-Glycans are added to proteins in the ER and undergo trimming and maturation in the ER and in the Golgi apparatus, where the properly folded proteins are sorted toward various cellular destinations (*Taniguchi and Aebi, 2015*). However, proteins that fail to fold properly are recognized, retrotranslocated from the ER into the cytosol and undergo proteasomal degradation through a process called ER-associated degradation (ERAD) (*Smith et al., 2011*; *Brodsky, 2012*). The yeast NGLY1 homolog (PNG1) is part of the ERAD mechanism (*Hirayama et al., 2015*), and NGLY1 is proposed to contribute to ERAD by removing *N*-glycans from misfolded proteins after their retrotranslocation, thereby promoting their degradation

(*Suzuki et al., 2016*). A fraction of NGLY1 molecules are associated with the ER (*Suzuki et al., 1997a*; *Katiyar et al., 2004*), likely through protein-protein interaction with valosin containing protein (VCP; also called p97) (*Ye et al., 2001*; *Li et al., 2006*). However, the functional significance of NGLY1's ER recruitment is not known, and the temporal relationship between retrotranslocation and deglycosylation has not been determined.

During *Drosophila* embryonic development, signaling by a bone morphogenetic protein (BMP) called Decapentaplegic (Dpp) is responsible for the specification of two regions in the middle part of the intestine (midgut), the gastric caeca region and the acid zone (*Panganiban et al., 1990*; *Newfeld et al., 1996*; *Dubreuil, 2004*). Dpp is first expressed in narrow bands in parasegments 3 (PS3) and PS7 of the embryonic visceral mesoderm (VM). Dpp then uses a paracrine/autocrine loop to sustain high levels of its own expression in the VM. As Dpp level increases in the PS3 and PS7 regions of the VM, it activates BMP signaling in the neighboring endoderm and induces the formation of gastric caeca and acid zone regions of the midgut (*Panganiban et al., 1990*; *Hursh et al., 1993*; *Bienz, 1997*; *Galeone et al., 2017*). We have previously shown that the *Drosophila* homolog of NGLY1 (PNGase-like or Pngl) is required in the VM to promote Dpp autoactivation in this tissue and consequently BMP signaling in the midgut endoderm (*Galeone et al., 2017*). However, the direct target of Pngl in the BMP pathway and the mechanism for the regulation of BMP signaling by Pngl are not known. Moreover, given the tissue-specific BMP defects observed in *Drosophila Pngl* mutants (*Galeone et al., 2017*), it remained to be seen whether NGLY1 regulates BMP signaling in mammals as well.

Here, we provide mechanistic evidence of the regulation of BMP pathway by Pngl/NGLY1 in flies and mammals. Our data indicate that Pngl/NGLY1 promotes Dpp/BMP4 signaling by removing *N*-glycans from misfolded Dpp/BMP4. Analysis of $Ngly1^{-/-}$ mouse embryos shows developmental abnormalities accompanied by a severe decrease in the expression of the BMP effector pSMAD1/5 in the heart and brain. Unexpectedly, our data suggest that BMP4 deglycosylation is specifically mediated by NGLY1 molecules recruited to the ER membrane, not the free cytosolic pool of NGLY1. Moreover, loss of NGLY1 or impaired recruitment of NGLY1 to the ER results in the accumulation of misfolded BMP4 in the ER not in the cytosol, strongly suggesting that deglycosylation of BMP4 by NGLY1 occurs before BMP4 is fully retrotranslocated from the ER. Our studies identify a new biologically relevant target of deglycosylation by NGLY1 and challenge the current assumption about the order of events during glycoprotein ERAD.

## Results

### *Drosophila* Dpp is a direct target of Pngl/NGLY1 in vivo

BMP ligands have a number of *N*-glycosylation sites, many of which are experimentally verified (*Gelbart, 1989*; *Groppe et al., 1998*; *Tauscher et al., 2016*). Therefore, one potential mechanism to explain the impaired Dpp signaling in *Pngl* mutants is that removal of *N*-glycans from Dpp promotes Dpp signaling in the VM. To test this hypothesis, we first overexpressed a GFP-tagged, functional version of Dpp (*Teleman and Cohen, 2000*) in the mesoderm using the GAL4/UAS system (*Brand and Perrimon, 1993*) and asked whether RNAi-mediated *Pngl* knock-down affects the migration of Dpp-GFP in western blots. In control embryos, immunoblotting with an anti-GFP antibody recognized a number of bands, corresponding in size to full-length Dpp-GFP and its cleavage products (*Figure 1A*). Upon *Pngl* knock-down, one of the Dpp-GFP bands shifted upwards, consistent with the presence of *N*-glycans that would normally be removed by Pngl. Incubation of the protein extract from these animals with PNGase F, an enzyme that removes *N*-glycans from *N*-glycoproteins (*Freeze and Kranz, 2010*), restored Dpp-GFP migration in mesodermal *Pngl* knock-down larvae (*Figure 1A*). None of the 13 asparagine residues present in GFP conform to the *N*-glycosylation minimal consensus sequence to which the majority of *N*-glycans are attached (N-X-S/T, where S is serine, T is threonine and X can be any amino acid other than proline) (*Taniguchi and Aebi, 2015*). These data provide strong evidence that Pngl deglycosylates a significant fraction of Dpp in *Drosophila* embryos.

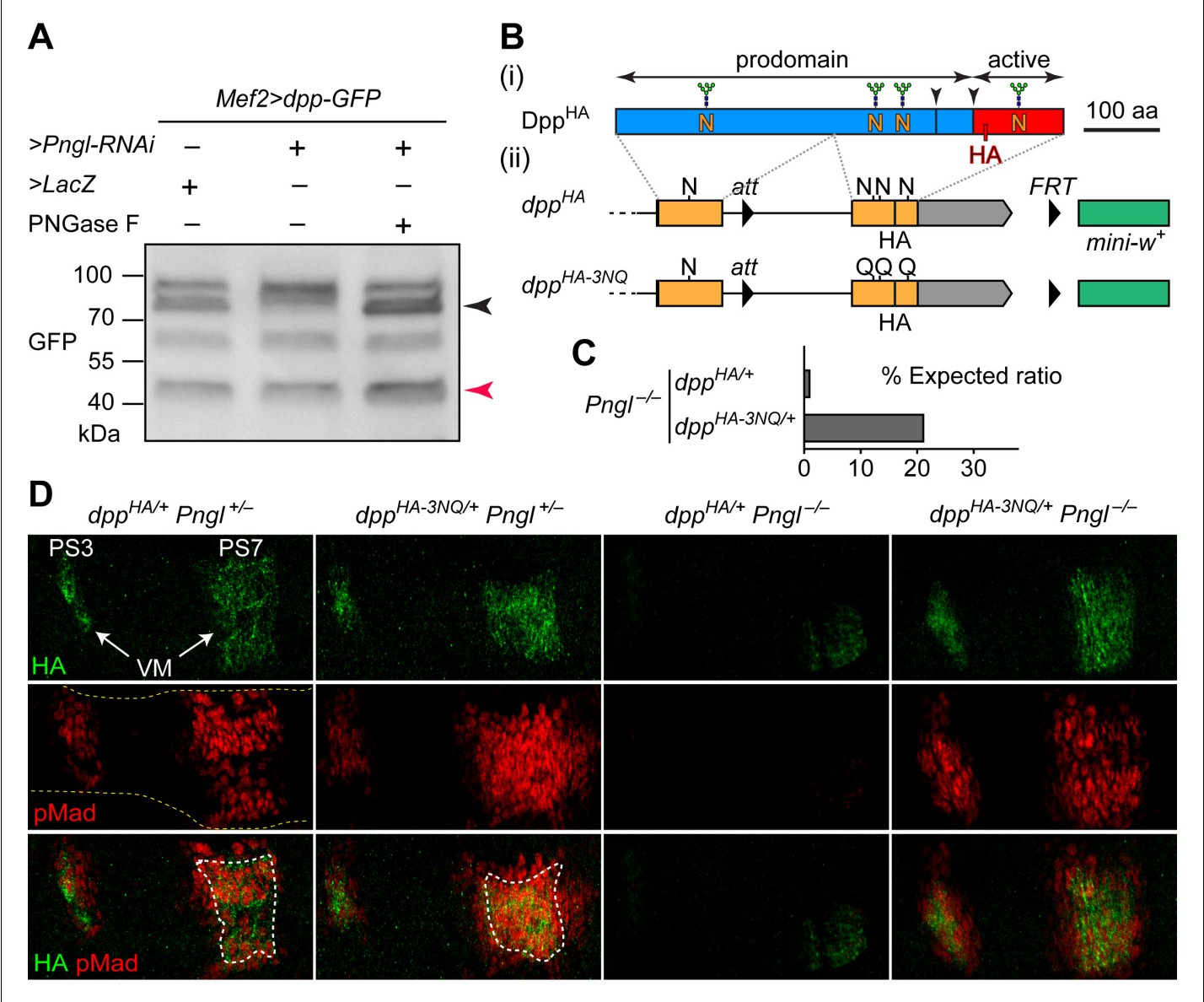

**Figure 1.** Deglycosylation of Dpp by Pngl is essential for BMP signaling during *Drosophila* midgut development. (**A**) Western blot with α-GFP on protein lysates from embryos of indicated genotypes. The shift in band size upon treatment with PNGase F (black arrowhead) shows that full-length Dpp-GFP retains *N*-glycans upon RNAi-mediated *Pngl* knock-down in the embryonic mesoderm. The mature Dpp-GFP is indicated by the red arrowhead. (**B**) (**i**) Schematic representation of Dpp$^{HA}$, which contains an HA tag in the active domain (red box). (**ii**) Schematic representation of *dpp*$^{HA}$ knock-in allele and its mutant version *dpp*$^{HA-3NQ}$, in which three of Dpp's four *N*-glycosylation sites are ablated by N-to-Q mutations. (**C**) Eclosion tests of *Pngl*$^{-/-}$ flies harboring a copy of *dpp*$^{HA}$ (1.3% of the expected Mendelian ratio, n = 153 total progeny scored) or *dpp*$^{HA-3NQ}$ (20.8% of the expected Mendelian ratio, n = 158 total progeny scored). One copy of *dpp*$^{HA-3NQ}$ partially rescues the lethality of *Pngl* mutant flies. (**D**) Immunofluorescence staining of parasegment 3 (PS3) to PS7 region of embryonic midguts (marked by the dashed yellow lines) of animals with the indicated genotypes. Dpp is expressed in PS3 and PS7 of the embryonic visceral mesoderm (VM) and induces BMP signaling (evidenced by pMad expression) in PS3 and PS7, both in the VM (paracrine/autocrine) and in the endoderm. pMad staining (red) indicates that one copy of *dpp*$^{HA-3NQ}$, but not *dpp*$^{HA}$, restores the Dpp signaling during *Pngl*$^{-/-}$ midgut development. n = 5 for each genotype. Scale bar, 50 μm.

The online version of this article includes the following figure supplement(s) for figure 1:

**Figure supplement 1.** The *dpp*$^{HA-3NQ}$ knock-in allele does not impair Dpp signaling in the midgut.

## Removal of Dpp *N*-glycans by Pngl/NGLY1 is essential for BMP signaling in *Drosophila* embryonic midgut

Given the above observations, we sought to examine whether removing *N*-glycans from Dpp by Pngl affects BMP signaling in the embryonic midgut. To this end, we took advantage of two transposable elements flanking the second (last) coding exon of *dpp* and generated a knock-in, HA-tagged allele of *dpp* in which the three *N*-glycosylation sites located in this exon are replaced by N-to-Q mutations (*Figure 1B*, *dpp*$^{HA-3NQ}$). If Pngl promotes BMP signaling in the VM by directly removing *N*-glycans from Dpp (as opposed to affecting other potential targets and indirectly regulating the function of Dpp), Dpp$^{HA-3NQ}$ should not depend on Pngl for signaling in the VM. To test this hypothesis, we set crosses leading to the generation of *Pngl*$^{-/-}$ animals in which one of the two endogenous copies of *dpp* is replaced with *dpp*$^{HA}$ or *dpp*$^{HA-3NQ}$. In agreement with our previous report on *Pngl*$^{-/-}$ animals (*Galeone et al., 2017*), only a small fraction (1.3%) of *dpp*$^{HA/+}$ *Pngl*$^{-/-}$ animals reached adulthood (*Figure 1C*). However, ~21% of *dpp*$^{HA-3NQ/+}$ *Pngl*$^{-/-}$ animals reached adulthood (*Figure 1C*). We have previously reported that loss of BMP signaling in the midgut is only responsible for up to 30% of the lethality in *Pngl*$^{-/-}$ animals (*Galeone et al., 2017*). Therefore, the BMP-dependent component of *Pngl*$^{-/-}$ lethality is largely rescued by genetic removal of these three *N*-glycans from Dpp.

In control (*Pngl*$^{+/-}$) embryos, both *dpp*$^{HA}$ and *dpp*$^{HA-3NQ}$ drive high levels of Dpp expression in PS3 and PS7 (*Figure 1D*), similar to endogenous Dpp expression pattern (*Galeone et al., 2017*). To assess the level of BMP signaling, we performed antibody staining for phosphorylated Mothers against dpp (pMad), which is the transducer of BMP signaling in flies (*Newfeld et al., 1996*). The pMad expression domain is broader in PS7 of *dpp*$^{HA-3NQ/+}$ *Pngl*$^{+/-}$ embryos compared to *dpp*$^{HA/+}$ *Pngl*$^{+/-}$ embryos (*Figure 1D*), suggesting that the 3NQ mutations might enhance BMP signaling in PS7. Similar to *Pngl*$^{-/-}$ embryos (*Galeone et al., 2017*), *dpp*$^{HA/+}$ *Pngl*$^{-/-}$ embryos show a severe decrease in pMad expression accompanied by weak and narrow Dpp expression domains in PS3 and PS7 (*Figure 1D*). Strikingly, one copy of *dpp*$^{HA-3NQ}$ is sufficient to rescue Dpp and pMad expression in PS3 and PS7 of *Pngl*$^{-/-}$ embryos (*Figure 1D*). These observations indicate that Pngl regulates BMP signaling in the embryonic VM by removing these three *N*-glycans or a subset of them from Dpp, and that the N-to-D conversion of *N*-glycosylated amino acids upon Pngl-mediated deglycosylation is not essential for BMP signaling during *Drosophila* midgut development.

We also examined the effect of homozygosity for *dpp*$^{HA}$ and *dpp*$^{HA-3NQ}$ alleles on BMP signaling in embryos and on animal survival. Staining *dpp*$^{HA-3NQ/HA-3NQ}$ and control *dpp*$^{HA/HA}$ embryos for HA revealed a similar expression pattern for Dpp$^{HA}$ and Dpp$^{HA-3NQ}$ but was somewhat broader for Dpp$^{HA-3NQ}$ (*Figure 1—figure supplement 1*). While the overall pMad expression pattern was similar in these two genotypes, pMad staining in the PS3 and PS7 areas of the developing midgut was expanded in *dpp*$^{HA-3NQ/HA-3NQ}$ animals (*Figure 1—figure supplement 1*). These data indicate that mutating these three Dpp *N*-glycosylation sites does not impair Dpp signaling during embryogenesis, and suggest that the mutations might even enhance Dpp's range of signaling in the midgut region. The control *dpp*$^{HA/HA}$ animals reached adulthood at the expected Mendelian ratio (n = 225). However, *dpp*$^{HA-3NQ/HA-3NQ}$ animals did not reach adulthood (n = 97) and died by the second instar stage. While we do not know the reason for the lethality of *dpp*$^{HA-3NQ/HA-3NQ}$ animals, the HA and pMad staining data suggest that the lethality is not due to the loss of BMP signaling in the embryos.

## *Ngly1*-mutant mouse embryos show a severe decrease in BMP signaling in some contexts

NGLY1 deficiency patients exhibit global developmental delay, seizures, involuntary movements, chronic constipation, osteopenia, small hands and feet, lack of tears and other symptoms (*Need et al., 2012*; *Enns et al., 2014*; *Lam et al., 2017*). A loss-of-function mutation in the mouse *Ngly1* (*Ngly1*$^{tm1.1Tasuz}$) results in late embryonic lethality accompanied by ventricular septal defects (VSD) (*Fujihira et al., 2017*). To examine whether mammalian BMP signaling is affected by the loss of *Ngly1*, we performed pSMAD1/5 staining in E15.5 mouse embryos homozygous for a newly generated *Ngly1* null mutant (*Ngly1*$^{em4Lutzy}$; *Figure 2—figure supplement 1*) and sibling controls. We focused our analysis on the heart (because of the above-mentioned report [*Fujihira et al., 2017*]) and on the CNS (because of human patient phenotypes [*Need et al., 2012*; *Enns et al., 2014*; *Lam et al., 2017*]). All five *Ngly1*$^{-/-}$ mutant embryos examined in our studies had VSDs (not shown).

*Ngly1*⁺/⁺ (control) embryos showed elaborate myocardial trabeculae expressing high level of pSMAD1/5 (*Figure 2A*, arrowheads). We found an overall decrease in the heart size and a remarkable reduction of myocardial trabeculae in mutant embryos (*Figure 2B*). The remaining trabeculae showed a much weaker pSMAD1/5 staining compared to controls (*Figure 2B*, arrowhead). In the CNS, we found significant abnormalities in the 4th ventricle choroid plexus. In *Ngly1*⁺/⁺ E15.5

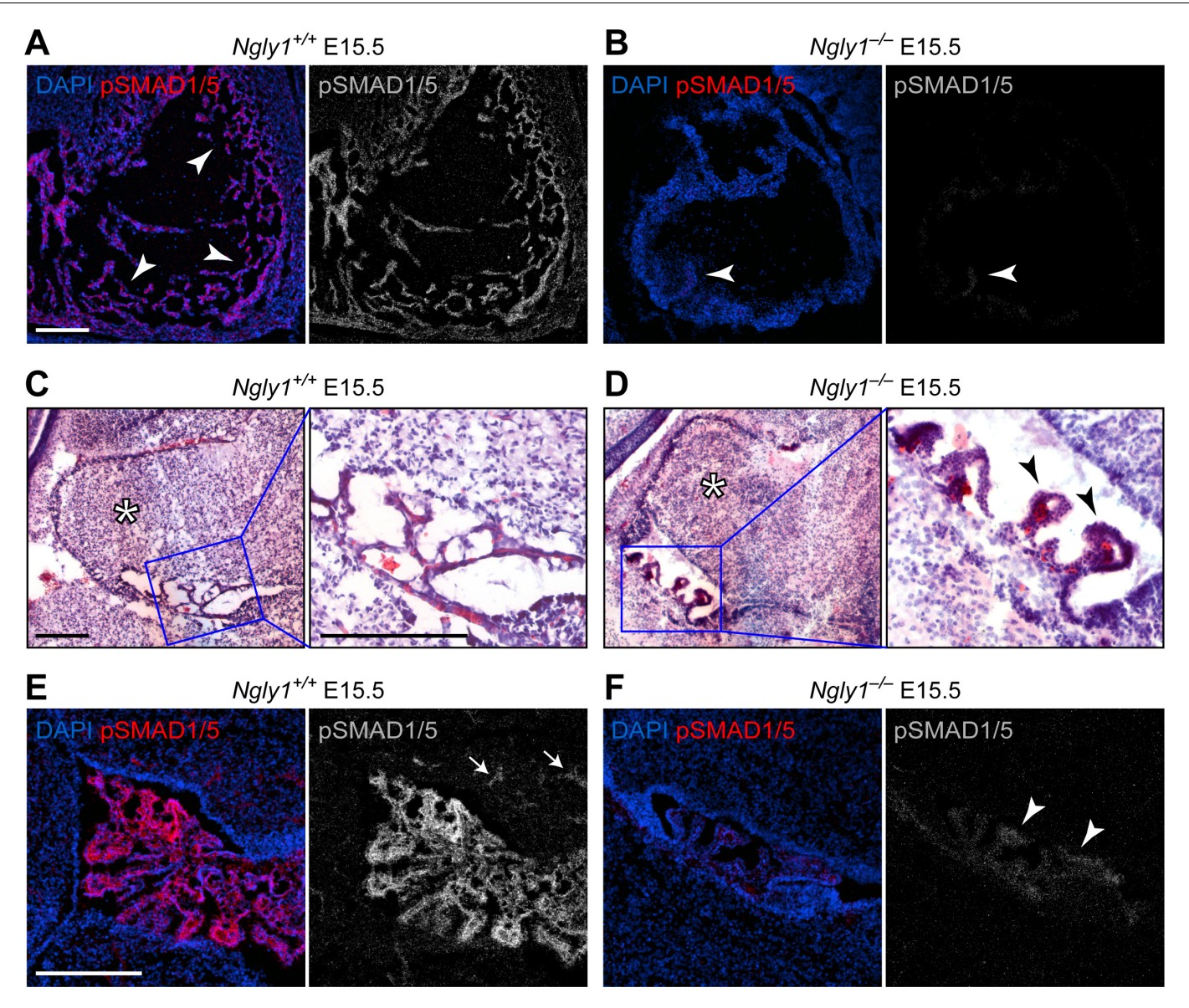

**Figure 2.** *Ngly1* is required for BMP signaling in embryonic heart and cerebellum. (**A,B**) Immunofluorescence staining in sagittal sections of the embryonic heart of control (*Ngly1*⁺/⁺) and *Ngly1*⁻/⁻ animals at E15.5. Nuclei (DAPI) are in blue and pSMAD1/5 in red. n = 5 for each genotype. pSMAD1/5 staining is severely reduced in mutant embryos (compare grey scale images), suggesting that loss of *Ngly1* leads to reduced BMP signaling in the heart. Arrowheads mark the myocardial trabeculae. (**C–F**) H&E (**C,D**) and immunofluorescence (**E,F**) staining in sagittal sections of embryonic cerebellum of control (*Ngly1*⁺/⁺) and *Ngly1*⁻/⁻ animals at E15.5. n = 5 for each genotype. Loss of *Ngly1* leads to significant morphological abnormalities in the 4th ventricle choroid plexus (close-up images from blue boxes) and severe reduction in pSMAD1/5 staining (in grey scale). Scale bars are 100 µm in A,B,E,F and 200 µm in c,d (both low magnification and close-up images). These data suggest that *Ngly1* has a role in BMP signaling in developing choroid plexus.

The online version of this article includes the following figure supplement(s) for figure 2:

**Figure supplement 1.** Schematic of the mouse *Ngly1* locus and the location of the 11 bp deletion in the *Ngly1*^em4Lutzy allele.

embryos, the choroid plexus consists of slender structures that express high levels of pSMAD1/5 and reach both sides of the ventricle (*Figure 2C and E*). However, *Ngly1⁻/⁻* embryos have a rudimentary, stump-like choroid plexus with a dramatic decrease in pSMAD1/5 expression (*Figure 2D and F*, arrowheads). pSMAD1/5 staining was also observed in control E15.5 cerebella (*Figure 2E*, arrowheads) but was significantly reduced in mutant cerebella (*Figure 2F*), although the morphological difference between control and mutant cerebella was less stark than that between control and mutant choroid plexus (*Figure 2C and D*, asterisks). BMP signaling, mediated by several ligands including BMP4, is essential for normal cardiovascular and choroid plexus development (*Hébert et al., 2002*; *Hébert et al., 2003*; *Morrell et al., 2016*), and BMP4 is sufficient to induce choroid plexus epithelial fate in neuroepithelial progenitors (*Lehtinen et al., 2013*). Therefore, these data indicate that similar to fly Pngl, mouse NGLY1 is required for BMP signaling in specific contexts and suggest a possible causative role between decreased BMP signaling and the observed phenotypes.

## NGLY1 promotes BMP4 signal-sending in mouse embryonic fibroblasts

To directly test the role of NGLY1 in mammalian BMP signaling, we performed staining and immunoblotting with an anti-pSMAD1/5 antibody in *Ngly1⁻/⁻* and control mouse embryonic fibroblasts (MEFs) (*Huang et al., 2015*). A low level of pSMAD1/5 expression was detected in wild-type MEFs, which did not show a statistically significant decrease in *Ngly1⁻/⁻* MEFs (*Figure 3A,C and D*). Transfection of control MEFs with a construct expressing double-tagged BMP4 (HA-tag in the prodomain and Myc-tag in the active domain) resulted in a 12-fold increase in BMP signaling (*Figure 3B–D*). However, BMP signaling in *Ngly1⁻/⁻* MEFs remained at the baseline level upon *Bmp4^HA-Myc* transfection (*Figure 3B–D*). Of note, treating control and *Ngly1⁻/⁻* cells with recombinant BMP4 resulted in similar levels of pSMAD1/5 expression (*Figure 3C,D*), indicating that NGLY1 is not essential for receiving the BMP4 signal. Together, these data indicate impaired BMP4 signal-sending in *Ngly1⁻/⁻* MEFs, similar to the impaired Dpp signal-sending observed in *Pngl⁻/⁻* fly VM (*Galeone et al., 2017*).

To determine whether inhibition of NGLY1 leads to impairment of endogenous BMP4 signaling in a mouse cell line, we used the preadipocyte 3T3-L1 cells line, which expresses *Bmp4* and depends on endogenous BMP4 signaling for pSMAD1/5 expression and adipocyte differentiation (*Suenaga et al., 2010*; *Suenaga et al., 2013*). As shown in *Figure 3—figure supplement 1*, treating 3T3-L1 cells with low and high concentrations of the NGLY1 inhibitor Z-VAD-fluoromethylketone (fmk) (*Misaghi et al., 2004*) resulted in a significant decrease in pSMAD1/5 levels in these cells, indicating reduced BMP signaling. Z-VAD-fmk is also a potent pan-caspase inhibitor, especially at high concentrations (*Misaghi et al., 2004*). To rule out the possibility that Z-VAD-fmk reduced BMP signaling in 3T3-L1 cells through its caspase inhibitory effect, we treated these cells with another broad spectrum caspase inhibitor, Q-VD-OPh, that does not inhibit NGLY1 (*Caserta et al., 2003*; *Misaghi et al., 2004*; *Tomlin et al., 2017*). As shown in *Figure 3—figure supplement 1*, 3T3-L1 cells treated with Q-VD-OPh did not exhibit any reduction in the level of pSMAD1/5. These data are in agreement with our *Drosophila* and mouse staining results and our MEF overexpression studies, and suggest that NGLY1 plays a key role in endogenous BMP4 signaling in some contexts.

## Loss of NGLY1 results in accumulation of BMP4 in the ER and upregulation of the ER stress markers

Our observations in MEFs and mouse embryos, combined with data establishing Dpp as a target of Pngl, prompted us to examine whether loss of *Ngly1* affects the glycosylation status and trafficking of BMP4. Double staining for HA and the ER marker KDEL showed that BMP4^HA-Myc only partially localizes to the ER in control MEFs (*Figure 4A*), likely reflecting the exocytic trafficking of BMP4 from ER to Golgi and beyond. In contrast, BMP4^HA-Myc almost fully colocalized with KDEL in *Ngly1⁻/⁻* MEFs (*Figure 4A*). This suggests that in the absence of NGLY1, BMP4 is trapped in the ER. The anti-KDEL antibody used in our studies is raised against the KDEL-containing C-terminal part of the GRP78/BiP and can recognize this protein in immunoblots (*Cai et al., 1998*; *Barra et al., 2017*). GRP78/Bip (official name: heat shock protein family A member 5, HSPA5) is a major ER luminal chaperone whose expression can be used to monitor ER stress (*Lee, 2005*). As shown in *Figure 4B and C*, the level of GRP78/BiP was significantly increased in control MEFs transfected with *Bmp4^HA-Myc*, suggesting some BMP4 misfolding in these cells. Untransfected *Ngly1⁻/⁻* MEFs also showed a similar

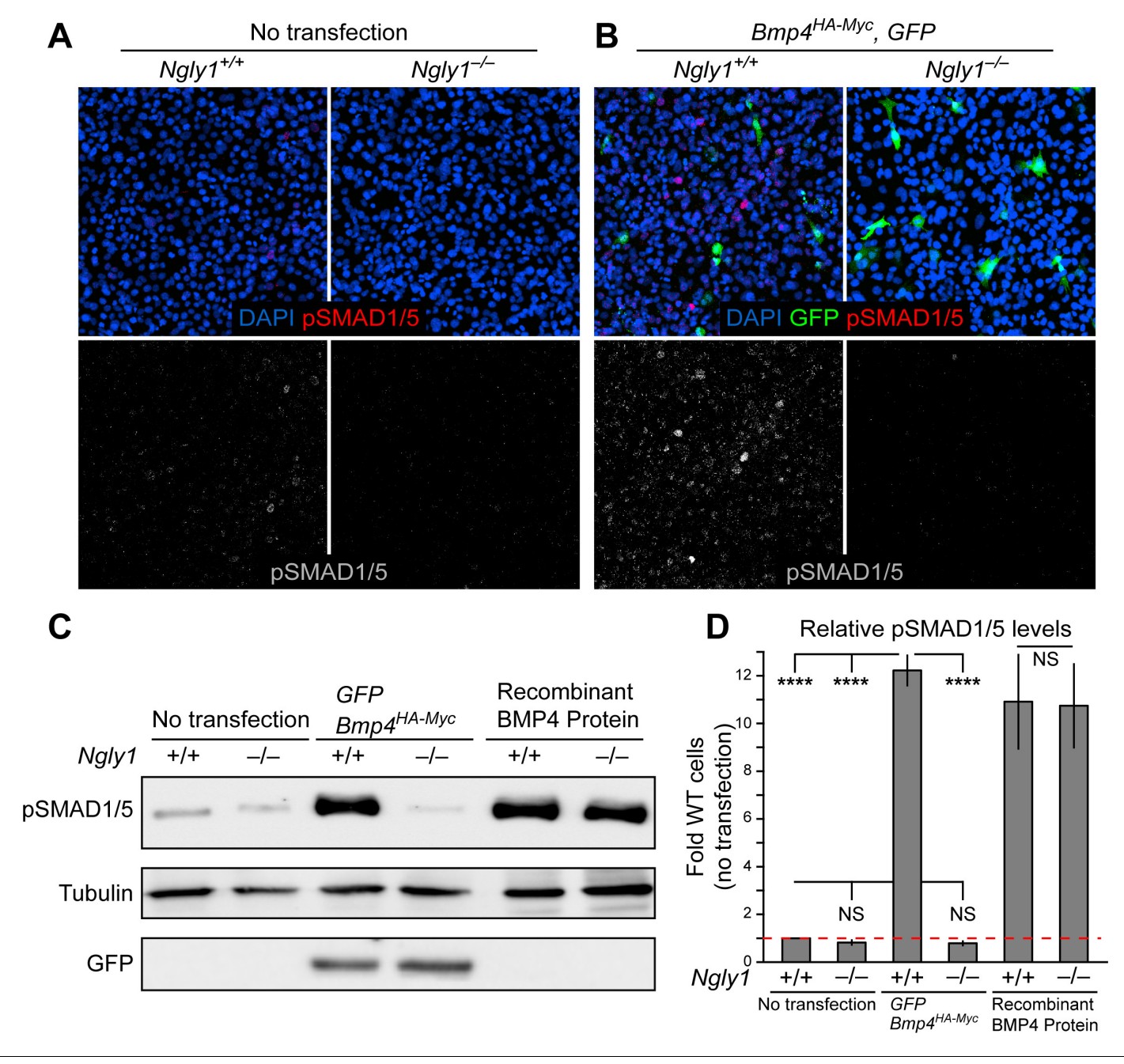

**Figure 3.** Loss of *Ngly1* leads to impaired BMP4 signal-sending in mouse embryonic fibroblasts. (**A**) Representative immunofluorescence staining of DAPI (blue) and pSMAD1/5 (red) in untransfected WT and *Ngly1* mutant (–/–) MEFs used as a baseline for pSMAD1/5 expression (grey scale). (**B**) WT and *Ngly1* mutant MEFs co-transfected with expression vectors for BMP4$^{HA-Myc}$ and GFP. GFP expression is used as control for transfection efficiency. pSMAD1/5 expression (grey scale) is used as readout of BMP signaling. Scale bars, 50 µm. n = 6 biologically independent samples. Loss of *Ngly1* severely reduces pSMAD1/5 levels. (**C**) Representative western blot of pSMAD1/5 from WT and *Ngly1* mutant MEF cell lysates. The first two lanes are without transfection (control); the middle two lanes are co-transfected with expression vectors for BMP4$^{HA-Myc}$ and GFP (for transfection efficiency); the last two lanes are treated with recombinant BMP4 in the culture media. (**D**) Quantification of the data shown in C. For each blot, the pSMAD1/5 expression was normalized to tubulin and the value of *Ngly1$^{+/+}$* cells without transfection was set as 1. n = 3 biologically independent samples. Two-way ANOVA with Tukey's multiple comparisons test was used for statistical analysis. Mean ± s.d. are shown. ****p<0.0001, NS, not significant. These data exhibit that BMP4 signal-sending, but not signal-receiving, is impaired in MEFs upon loss of *Ngly1*, similarly to the impaired Dpp signal-sending in *Drosophila* embryonic midguts (*Galeone et al., 2017*).

The online version of this article includes the following source data and figure supplement(s) for figure 3:

**Source data 1.** Raw data and statistical analysis for panel D.

*Figure 3 continued on next page*

*Figure 3 continued*

**Figure supplement 1.** Endogenous BMP signaling is impaired upon NGLY1 inhibition in 3T3-L1 cells.

**Figure supplement 1—source data 1.** Raw data and statistical analysis for the graph.

increase in GRP78/BiP expression (*Figure 4B and C*), suggesting some degree of ER stress in these cells (*Kozutsumi et al., 1988*; *Gülow et al., 2002*). Upon *Bmp4*$^{HA-Myc}$ transfection, *Ngly1*$^{-/-}$ cells showed further increase in GRP78/BiP level (*Figure 4B and C*), suggesting that expression of BMP4$^{HA-Myc}$ leads to additional ER stress in *Ngly1*$^{-/-}$ cells.

To provide further evidence for ER stress in *Ngly1*$^{-/-}$ cells especially upon BMP4 overexpression, we examined the levels of two additional ER stress markers: phosphorylated IRE1α (pIRE1α), which is an indicator of unfolded protein response activation (*Zhang and Kaufman, 2004*; *Korennykh et al., 2009*), and OS9, an ER lectin which is upregulated upon ER stress, selectively binds misfolded glycoproteins, and facilitates their transport to the retrotranslocation machinery (*Kim et al., 2005*; *Alcock and Swanton, 2009*; *Satoh et al., 2010*). The levels of pIRE1α and OS9 were moderately increased in *Ngly1*$^{-/-}$ cells, further suggesting that loss of *Ngly1* leads to ER stress, potentially due to misfolded glycoprotein accumulation (*Figure 4B and C*). Moreover, upon *Bmp4*$^{HA-Myc}$ transfection, the levels of pIRE1α and OS9 in *Ngly1*$^{-/-}$ cells increased to 2.5 to 3-fold of pIRE1α and OS9 in untransfected *Ngly1*$^{-/-}$ cells. Together with GRP78 accumulation, these data demonstrate that *Bmp4* transfection leads to significant ER stress in *Ngly1*$^{-/-}$ MEFs, potentially due to the accumulation of misfolded BMP4 in the ER.

## Removal of BMP4 *N*-glycans by NGLY1 promotes BMP4 signal-sending in MEFs

The full-length BMP proteins dimerize in the ER, but the cleavages that release the dimerized active domain of BMP ligands from the prodomains are thought to occur in the Golgi apparatus and/or further along the secretory pathway (*Figure 4—figure supplement 1*; *Nelsen and Christian, 2009*). As expected from reduced signaling activity by BMP4$^{HA-Myc}$ in the absence of NGLY1, conditioned media from *Ngly1*$^{-/-}$ MEFs showed a severe decrease in the level of BMP4$^{Myc}$ active domain secreted by *Ngly1*$^{-/-}$ cells compared to control MEFs (*Figure 4D*). *Ngly1*$^{-/-}$ cells accumulated a band corresponding to the full-length BMP4$^{HA-Myc}$ that migrated slower than the full-length BMP4$^{HA-Myc}$ from control cells in SDS-PAGE gels, compatible with retention of *N*-glycans upon loss of NGLY1 (*Figure 4D*). Upon digestion of the cell lysates with PNGase F and Endo H enzymes (*Freeze and Kranz, 2010*), the BMP4$^{HA-Myc}$ band migrated similarly in *Ngly1*$^{+/+}$ and *Ngly1*$^{-/-}$ cells, indicating that BMP4 retains ER-type (high mannose) *N*-glycans in *Ngly1*$^{-/-}$ cells (*Figure 4D*). Moreover, a mutant version of BMP4$^{HA-Myc}$ harboring N-to-Q mutations in all four *N*-glycosylation sites (BMP4$^{HA-Myc-4NQ}$) was secreted efficiently by MEFs and was able to induce robust BMP signaling (*Figure 4E*). These observations indicate that in *Ngly1*$^{-/-}$ MEFs, BMP4$^{HA-Myc}$ accumulates in the ER and retains ER-type *N*-glycans. The data further support the notion that like Dpp, removal of BMP4 *N*-glycan(s) by NGLY1 is required for BMP4 signaling.

## BMP4/Dpp deglycosylation is specifically mediated by NGLY1 molecules recruited to the ER

The current model for the role of NGLY1 in ERAD posits that NGLY1 deglycosylates misfolded proteins in the cytosol after their retrotranslocation from the ER to facilitate proteasomal degradation (*Suzuki et al., 2016*) and predicts that loss of NGLY1 should result in the accumulation of its targets in the cytosol. Given the accumulation of glycosylated BMP4 in the ER of *Ngly1*$^{-/-}$ MEFs, we sought to reevaluate this model. NGLY1 is likely recruited from the cytosol to the ER membrane via its association with valosin containing protein (VCP; also called p97) (*Li et al., 2006*), an ATPase which is recruited to the ER upon the accumulation of misfolded proteins in the ER (*Kondratyev et al., 2007*) and plays a critical role in retrotranslocation of misfolded proteins from the ER to cytosol (*Ye et al., 2001*). Staining of *Ngly1*$^{-/-}$ MEFs transfected with an expression vector for a V5-tagged version of wild-type human NGLY1 (NGLY1$^{V5-WT}$) with anti-V5 and anti-KDEL antibodies did not show a clear association between NGLY1 and ER (*Figure 5A*). However, co-transfection of *Bmp4*$^{HA-Myc}$ with *NGLY1*$^{V5-WT}$ resulted in a significant overlap between V5 and KDEL signals (*Figure 5A and*

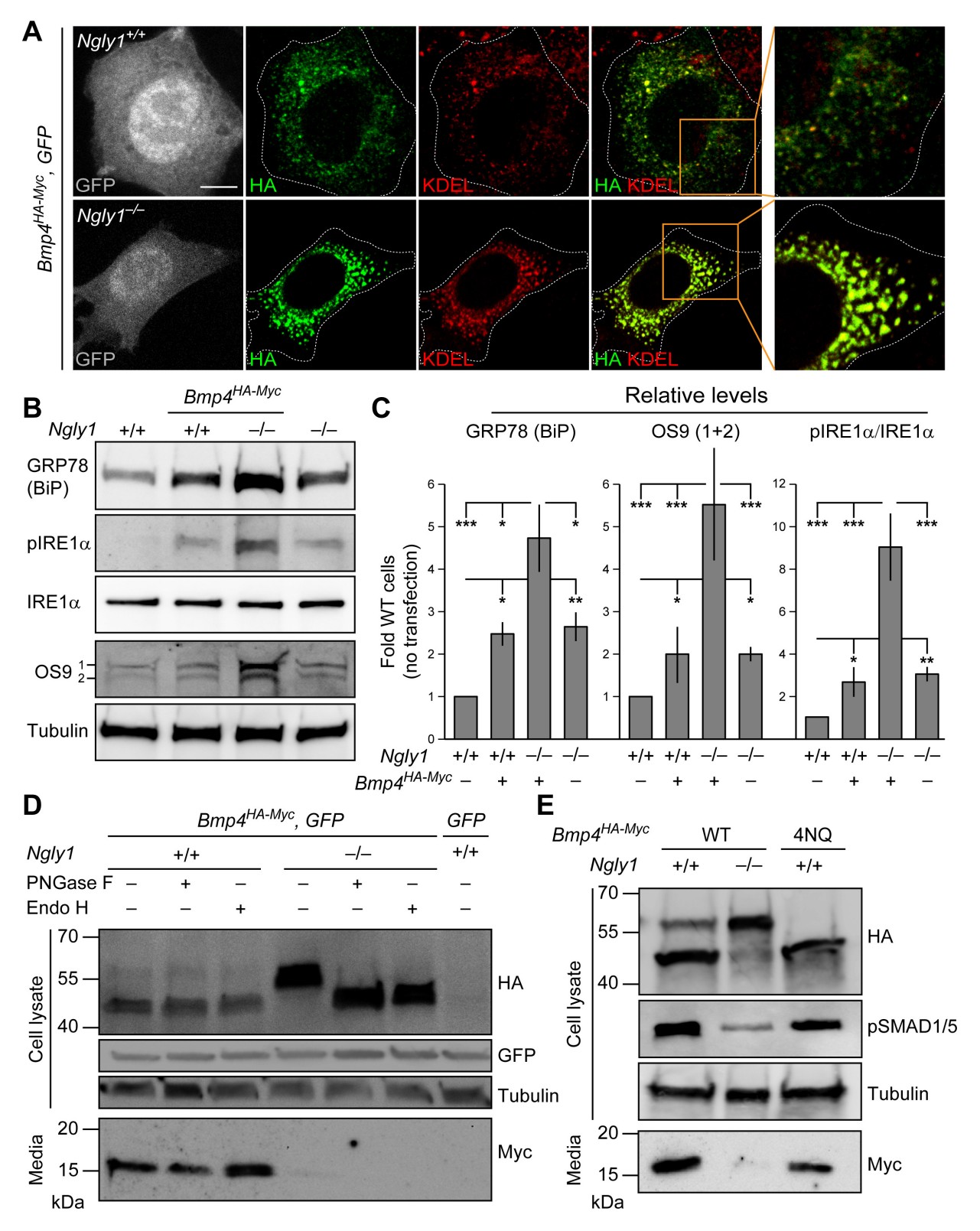

**Figure 4.** In *Ngly1* mutant MEFs, BMP4 retains *N*-glycans and is trapped in the ER. (**A**) Confocal sections of WT and *Ngly1* mutant MEFs co-transfected with *Bmp4*<sup>HA-Myc</sup> and *GFP* (for transfection efficiency) and stained for HA (to mark BMP4<sup>HA-Myc</sup>, green) and the ER marker KDEL (red). HA-tagged BMP4 almost fully colocalizes with KDEL/GRP78 in *Ngly1* mutant cells, suggesting impaired exocytic trafficking. Scale bars, 10 μm. n = 8 biologically independent samples. (**B,C**) Western blot with α-KDEL antibody, which marks GRP78 (BiP), and α-pIRE1α, α-IRE1α, and α-OS9 antibodies on protein
*Figure 4 continued on next page*

*Figure 4 continued*

lysates from WT and *Ngly1* mutant MEFs with or without *Bmp4$^{HA-Myc}$* transfection. ER stress is induced in *Ngly1$^{-/-}$* MEFs and is significantly enhanced upon *Bmp4* transfection, likely due to the accumulation of misfolded BMP4. A representative image of three independent blots is shown. For each blot, the protein expression was normalized to tubulin and the value of *Ngly1$^{+/+}$* cells without transfection was set as 1. Two-way ANOVA with Tukey's multiple comparisons test was used for statistical analysis. Mean ± s.d. is shown. *$p<0.05$; **$p<0.01$; ***$p<0.001$; NS, not significant. (D) Western blot for HA and Myc on cell lysates and media from WT and *Ngly1* mutant MEFs transfected with expression vectors for BMP4$^{HA-Myc}$ and GFP or GFP only (last lane). Protein extracts were treated with PNGase and Endo H for glycosylation profiling. Note the accumulation of a slow-migrating HA$^+$ band in *Ngly1$^{-/-}$* MEFs which returns to normal size upon treatment with both enzymes, indicating the retention of *N*-glycans on BMP4 upon loss of *Ngly1*. Note also the severe decrease in the level of active BMP4 (Myc blot) secreted by *Ngly1$^{-/-}$* MEFs into the culture media. (E) Western blot for HA, pSMAD1/5 and Myc on cell lysates and media from WT MEFs transfected with an expression vector for BMP4$^{HA-Myc-4NQ}$ compared with WT and *Ngly1$^{-/-}$* MEFs expressing wild-type BMP4$^{HA-Myc}$. n = 3 independent biological samples. These data indicate that *N*-glycans on BMP4 are not necessary for its secretion (Myc) and signaling (pSMAD1/5) and that NGLY1 promotes BMP4 signaling by removing one or more *N*-glycans from BMP4, similar to their *Drosophila* counterparts.

The online version of this article includes the following source data and figure supplement(s) for figure 4:

**Source data 1.** Raw data and statistical analysis for panel C.
**Figure supplement 1.** BMP4 processing in sending cells.

---

*B*), suggesting that NGLY1 is recruited to the ER upon ER stress, similar to VCP (*Kondratyev et al., 2007*). Next, we generated two mutant versions of human NGLY1 (N41P and G79A/F80A) previously shown to abolish NGLY1-VCP interaction (*Li et al., 2006*). As shown in *Figure 5A*, ER recruitment of NGLY1 was abolished in both mutant versions. These observations support the notion that upon ER stress, NGLY1 is recruited to the ER, potentially through its interaction with VCP.

We next asked whether ER recruitment of NGLY1 is required for BMP4 signaling. Overexpression of NGLY1$^{V5-WT}$ restored BMP4 deglycosylation and signaling in *Ngly1$^{-/-}$* MEFs (*Figure 6A*). However, NGLY1$^{V5-N41P}$ and NGLY1$^{V5-G79A/F80A}$ failed to promote BMP4 deglycosylation and signaling in these cells despite their comparable expression levels with wild-type NGLY1 (*Figure 6A*). Since N41, G79 and F80 are conserved between human NGLY1 and fly Pngl (*Li et al., 2006*), we also examined the effect of N41P and G79F/F80 mutations on the ability of human NGLY1 to rescue the loss of Dpp signaling in *Pngl$^{-/-}$ Drosophila* midgut. Transgenic expression of human NGLY1$^{WT}$ in the mesoderm fully rescued BMP signaling in *Drosophila* embryonic midguts (*Figure 6B*). Staining in wing imaginal discs of *dpp-GAL4 UAS-NGLY1$^{V5}$* animals indicated that the transgenes used in our studies are able to drive comparable expression levels of WT and mutant NGLY1 proteins in flies (*Figure 6—figure supplement 1*). However, transgenic expression of NGLY1$^{N41P}$ and NGLY1$^{G79A/F80A}$ in the mesoderm by *Mef2-GAL4* did not rescue the loss of pMad staining in the VM of *Pngl$^{-/-}$Drosophila* embryos (*Figure 6B*). Of note, in vitro deglycosylation assays indicated robust enzymatic activity for recombinant NGLY1$^{N41P}$ and NGLY1$^{G79A/F80A}$ (*Figure 6C*). Together, these data suggest that recruitment of the NGLY1 molecules from the cytosol to the ER is an evolutionarily conserved step required for BMP4/Dpp deglycosylation and signaling.

## Retrotranslocation of misfolded BMP4 into the cytosol, but not its subsequent proteasomal degradation, is required for BMP4 signaling

It has previously been reported that for certain misfolded molecules, blocking proteasomal degradation results in the accumulation of the misfolded protein in the ER, suggesting that retrotranslocation and proteasomal degradation are linked for those substrates (*Chillarón and Haas, 2000*; *Saliba et al., 2002*). To examine whether this is the case for BMP4, we evaluated the effects of the proteasomal inhibitor bortezomib (BTZ) on BMP4$^{HA-Myc}$ trafficking and signaling in wild-type MEFs. As reported previously (*Tomlin et al., 2017*; *Yang et al., 2018*), BTZ treatment impaired proteasome function in these cells, as evidenced by the accumulation of ubiquitinated proteins (*Figure 6D*, FK1). Anti-HA western blotting showed that BTZ treatment results in the accumulation of a band corresponding in size to deglycosylated BMP4$^{HA-Myc}$ (*Figure 6D and E*). This observation suggests that the NGLY1-mediated deglycosylation of BMP4 is normally followed by its proteasomal degradation. However, BTZ treatment of wild-type MEFs expressing BMP4$^{HA-Myc}$ did not impair BMP4 signaling, as evidenced by pSMAD1/5 expression (*Figure 6D*). These observations indicate that the critical NGLY1-dependent step in BMP4 signaling is the retrotranslocation of misfolded ligands from the ER, not their proteasomal degradation.

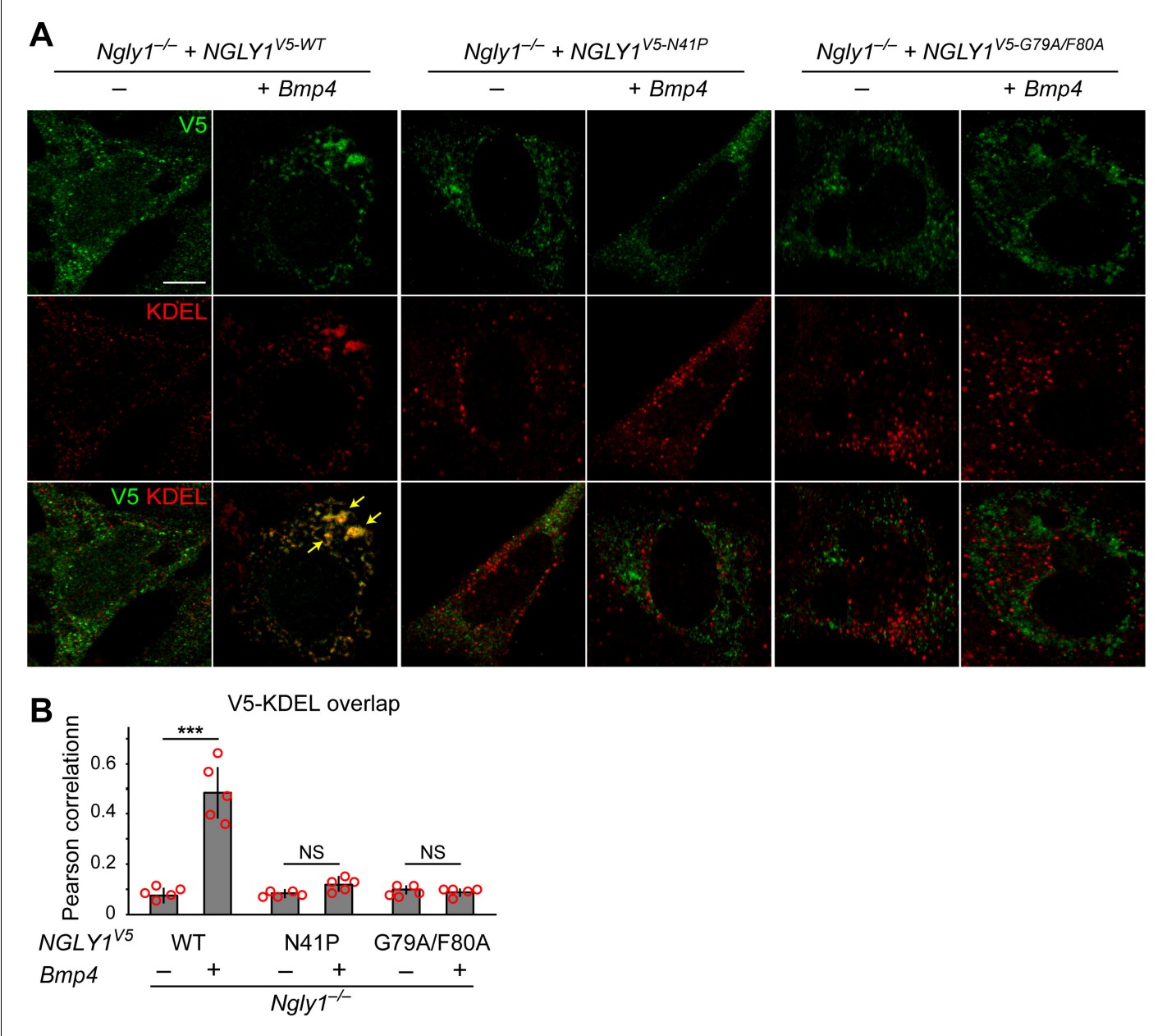

**Figure 5.** Mutations in VCP-binding sites abolish the recruitment of NGLY1 to ER. (**A**) Confocal sections of *Ngly1*$^{-/-}$ MEFs transfected with expression vectors for wild-type or VCP-binding mutant versions of human NGLY1 tagged with V5 and stained with V5 (green) and KDEL (ER marker, red). Note the ER recruitment of wild-type NGLY1 upon *Bmp4* transfection, which is likely due to the accumulation of misfolded BMP4 in the ER. Mutant NGLY1 proteins do not show ER recruitment. (**B**) Analysis of relative fluorescence overlap of V5-tagged NGLY1 (green) and KDEL signals (red) in A. The signal overlap is quantified by Pearson correlation analysis of five images from three independent experiments. Scale bars, 10 μm. Mean ± s.d. is shown. ***p=0.00011; NS, not significant. These results show that mutations in VCP-binding sites lead to a failure of the NGLY1 recruitment to ER.

The online version of this article includes the following source data for figure 5:

**Source data 1.** Raw data and statistical analysis for panel B.

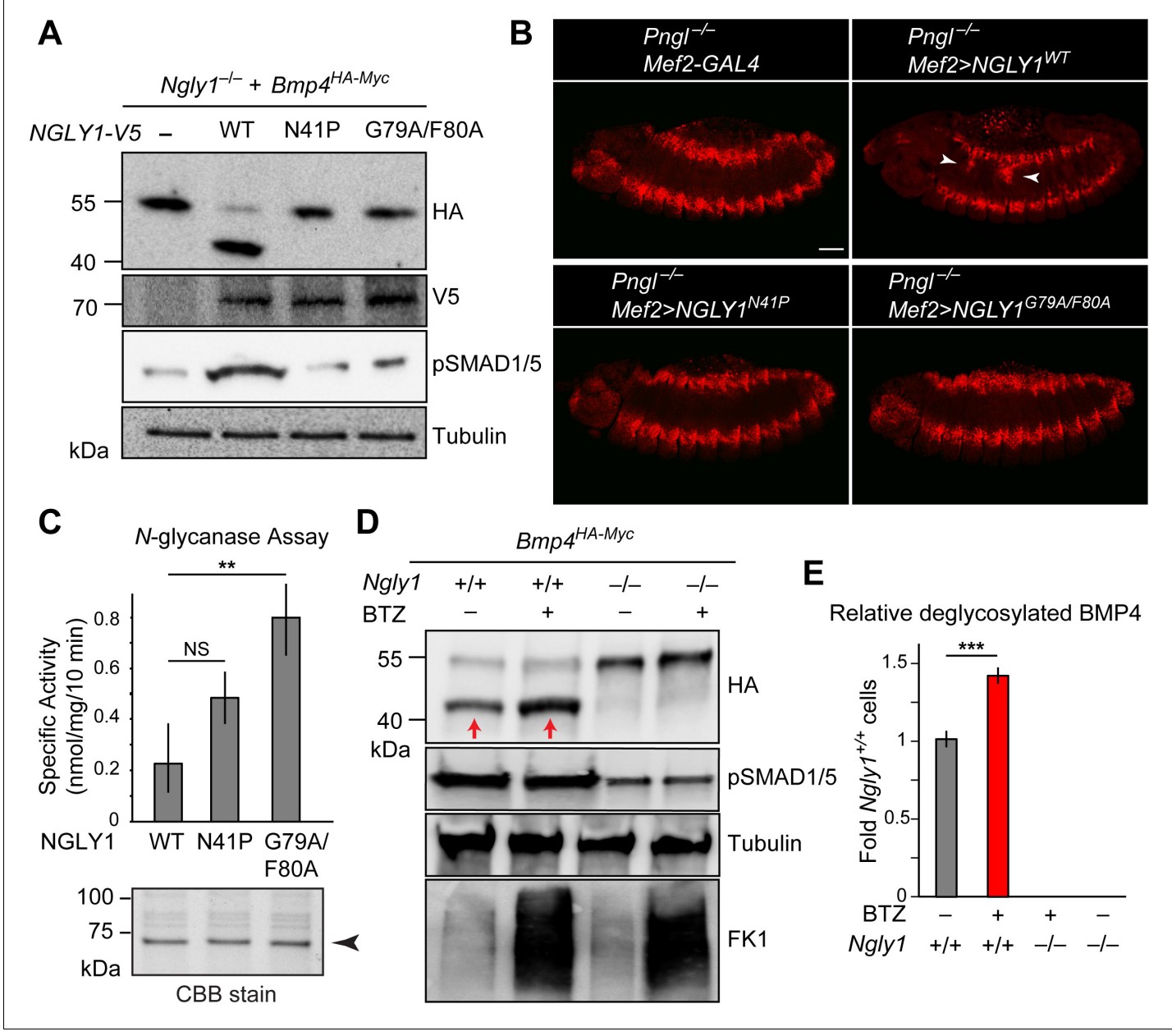

**Figure 6.** ER recruitment of NGLY1 is required for BMP4 deglycosylation and signaling in MEFs and Dpp signaling in *Drosophila* embryonic midgut. (**A**) Western blots with V5, HA and pSMAD1/5 antibodies on cell lysates from *Ngly1*[−/−] cells expressing BMP4[HA-Myc] alone or with human NGLY1[V5-His6] (wild-type or VCP-binding mutants). Wild-type NGLY1 efficiently deglycosylates BMP4 (downward shift of the HA[+] band) and induces robust BMP signaling (pSMAD1/5), but NGLY1 with mutations in its VCP-binding sites fail to deglycosylate BMP4 and induce BMP signaling. (**B**) Projection views of confocal image stacks of stage 14 embryos with indicated genotypes stained for pMad. Expression of wild-type human NGLY1 in mesoderm rescues BMP signaling in the midgut region (arrowheads), but the VCP-binding mutant versions of human NGLY1 do not rescue this phenotype. (**C**) Coomassie Brilliant Blue (CBB) stain of recombinant wild-type and VCP binding mutant (N41P and G79A/F80A) NGLY1[V5-His6] proteins generated in wheat germ cell-free system and deglycosylation activity assays on these proteins. Note that the VCP-binding mutants are stably expressed (arrowhead) and do not decrease NGLY1's enzymatic activity. One-way ANOVA with Tukey's multiple comparisons test was used. Mean ± s.d. **p<0.01; NS, not significant. (**D**) Western blot of HA-tagged BMP4 from cell lysates of WT and *Ngly1*[−/−] MEFs treated with bortezomib (BTZ, 10 nM for 6 hr). (**E**) Quantification of deglycosylated HA-tagged BMP4 expression. n = 3 biologically independent samples. Two-way ANOVA test with Tukey's multiple comparisons test was used for statistical analysis. Mean ± s.d. ***p<0.001. These results indicate that deglycosylated BMP4 is degraded by proteasome as result of ER-to-cytosol retrotranslocation. The proteasome impairment of BMP4 degradation does not block BMP signaling.

The online version of this article includes the following source data and figure supplement(s) for figure 6:

**Source data 1.** Raw data and statistical analysis for panels C and E.

*Figure 6 continued on next page*

*Figure 6 continued*

**Figure supplement 1.** Transgenic expression of human *NGLY1* in *Drosophila* wing discs.

## Misfolded BMP4 molecules bind an ER lectin and are substrate for ERAD

Our data strongly suggest that BMP4 is an ERAD substrate and that misfolded BMP4 molecules accumulate in the ER upon loss of NGLY1. To confirm these findings, we performed two additional experiments. It has been reported that the ER lectin OS9 accumulates upon ER stress and forms non-covalent complexes with misfolded glycoproteins to keep them in the ER and transport them to the retrotranslocation machinery (*Bernasconi et al., 2008*; *Christianson et al., 2008*). We reasoned that if the accumulated BMP4 molecules in $Ngly1^{-/-}$ MEFs are misfolded, they should interact with OS9. Indeed, co-immunoprecipitation (co-IP) experiments indicated that BMP4$^{HA-Myc}$ binds endogenous OS9 (*Figure 7A*), which accumulates in $Ngly1^{-/-}$ cells (*Figure 4B*). Treating $Ngly1^{-/-}$ MEFs with DTT also induced OS9 expression (*Figure 7A*), in agreement with induction of ER protein misfolding by DTT (*Lai et al., 2010*). However, the anti-HA antibody did not IP OS9 in DTT-treated $Ngly1^{-/-}$ MEFs, indicating that the observed IP of OS9 is specifically mediated by BMP4$^{HA-Myc}$. These data provide additional evidence that misfolded BMP4 molecules accumulate in the ER of $Ngly1^{-/-}$ MEFs. Importantly, co-transfection of $NGLY1^{V5-WT}$ with $Bmp4^{HA-Myc}$ into $Ngly1^{-/-}$ MEFs greatly reduced the level of OS9 in these cells, parallel to a rescue of BMP4 deglycosylation (*Figure 7A*, OS9 and HA panels). However, co-transfection of $NGLY1^{V5-N41P}$ or $NGLY1^{V5-G79A/F80}$, which impair the NGLY1-VCP interaction and ER recruitment of NGLY1, failed to reduce OS9 levels and showed a consistent BMP4-OS9 IP. These results indicate that loss of $Ngly1$ leads to the accumulation of misfolded BMP4 in the ER, and that interaction of NGLY1 with VCP is critical for retrotranslocation of misfolded BMP4 from the ER into the cytosol.

VCP is a critical component of ERAD (*Ye et al., 2001*; *Rabinovich et al., 2002*). Therefore, to directly examine whether misfolded BMP4 is an ERAD substrate, we transfected control ($Ngly1^{+/+}$) MEFs with $Bmp4^{HA-Myc}$ and treated them with the highly potent and selective VCP inhibitor NMS-873 (*Magnaghi et al., 2013*), BTZ, or both. In agreement with the data shown in *Figures 4* and *6*, BMP4$^{HA-Myc}$ was mostly deglycosylated in control cells (*Figure 7B*). NMS-873 treatment led to the accumulation of glycosylated BMP4$^{HA-Myc}$ in MEFs (*Figure 7B*, upper band in HA panel), accompanied by a reduction in the deglycosylated BMP4$^{HA-Myc}$ (*Figure 7B*, lower band in HA panel) and an increase in the level of GRP78/BiP. These observations support the conclusion that when the function of VCP is compromised, misfolded BMP4 fails to be deglycosylated by NGLY1 and remains in the ER, thereby inducing ER stress. Combination of NMS-873 and BTZ resulted in the accumulation of deglycosylated BMP4$^{HA-Myc}$ and an enhanced accumulation of glycosylated BMP4$^{HA-Myc}$ and GRP78/BiP compared to cells treated with NMS-873 alone. Together, these data provide compelling evidence that BMP4 is an ERAD substrate.

## ER recruitment of NGLY1 by VCP is not required for NFE2L1 activation

An unusual ER membrane protein whose ER luminal domain contains DNA-binding and transactivation domains (SKN-1 in *C. elegans*, NFE2L1 or NRF1 in mammals) has recently been shown to be a direct target of *N*-glycanase one in worms and mammalian cells (*Tomlin et al., 2017*; *Lehrbach et al., 2019*). In cells with normal proteasome function, NFE2L1 is retrotranslocated from the ER in a VCP-dependent manner and undergoes proteasomal degradation (*Steffen et al., 2010*; *Radhakrishnan et al., 2014*). However, upon impaired proteasomal function, SKN-1/NFE2L1 undergoes deglycosylation and N-to-D sequence editing by PNG-1/NGLY1 and cleavage by a conserved cytoplasmic protease, and enters the nucleus to activate the transcription of proteasomal components, a process known as proteasome bounce-back response (*Radhakrishnan et al., 2010*; *Lehrbach and Ruvkun, 2016*; *Tomlin et al., 2017*). To examine whether NFE2L1 deglycosylation is specifically mediated by the ER-associated NGLY1 similar to BMP4, we examined the ability of wild-type and VCP-binding-deficient mutant versions of human NGLY1 to restore the proteasomal bounce-back response in $Ngly1^{-/-}$ MEFs. In agreement with a previous report (*Tomlin et al., 2017*), treatment of $Ngly1^{+/+}$ MEFs with BTZ induced the expression of proteasomal genes *Psmb1*, *Psmb4* and *Psmb7* (*Figure 8A–C*), but in $Ngly1^{-/-}$ cells this response was completely lost (*Figure 8A–C*,

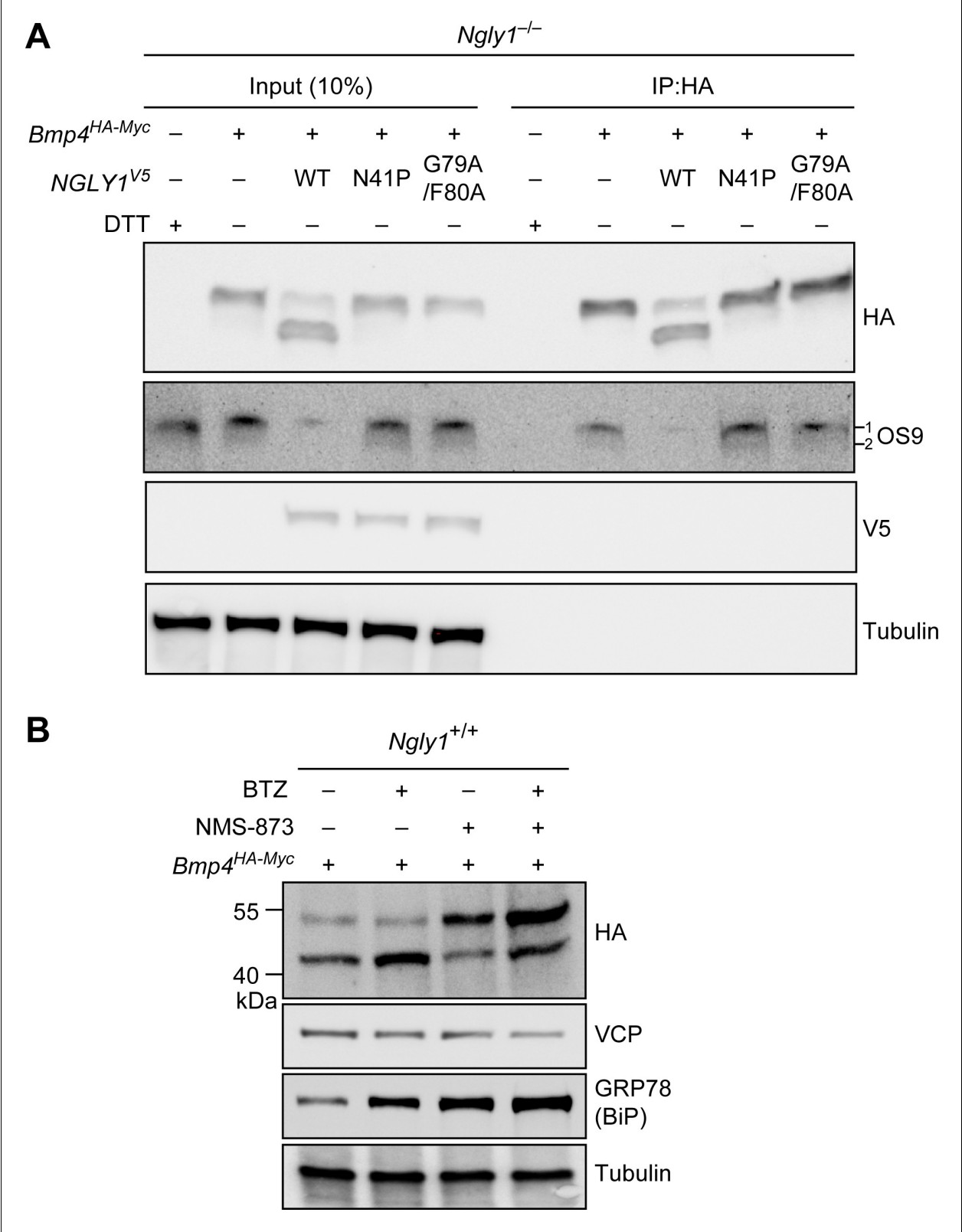

**Figure 7.** Misfolded BMP4 binds the ER lectin OS9 and is a substrate of ERAD. (**A**) Co-IP analysis of HA-tagged BMP4 and OS9 from $Ngly1^{-/-}$ MEFs. The lysates were immunoprecipitated with HA and analyzed by immunoblotting with α-HA and α-OS9 antibodies. Untransfected cells treated with 2 mM DTT were used as control. Input represents 10% of the total cell extract used for IP. Note that BMP4 binds OS9 when NGLY1 is missing or fails to be recruited to the ER. (**B**) Western blot of HA-tagged BMP4 from $Ngly1^{+/+}$ MEF cell lysates treated with NMS-873 (5 μM for 6 hr) and/or BTZ (10 nM

*Figure 7 continued on next page*

Figure 7 continued

for 6 hr). Note that upon NMS-873 treatment, glycosylated BMP4 accumulates in the cell and the level of ER chaperone GRP78 (BiP) is increased, suggesting that misfolded BMP4 accumulates in the ER due to impaired ERAD.

arrows). In fact, the expression level of two out of three proteasomal genes examined here was significantly reduced upon BTZ treatment (*Figure 8B and C*). Overexpression of human NGLY1$^{V5-WT}$ rescued this phenotype in *Ngly1$^{-/-}$* MEFs (*Figure 8A–C*). Notably, unlike BMP4 signaling, the proteasomal bounce-back was also restored by NGLY1$^{V5-N41P}$ and NGLY1$^{V5-G79A/F80A}$ in *Ngly1$^{-/-}$* MEFs (*Figure 8A–C*). These observations indicate that the deglycosylation/N-to-D editing of NFE2L1 *N*-glycosylation sites does not depend on NGLY1's ER recruitment and can be mediated by the cytosolic pool of NGLY1.

## Discussion

The prevailing model for the function of NGLY1 has been that it deglycosylates some misfolded *N*-glycoproteins after they are retrotranslocated from the ER into the cytosol and thereby promotes their proteasomal degradation (*Suzuki et al., 2016*). Our data indicate that an alternative model should be considered for the function of NGLY1 in BMP signaling. Specifically, we propose that upon accumulation of misfolded BMP4 in the ER, some NGLY1 molecules are recruited to the ER and that the VCP-mediated retrotranslocation of misfolded BMP4 molecules can only be efficiently completed upon deglycosylation by NGLY1 molecules recruited to the cytosolic surface of the ER (*Figure 9*). This model is supported by the following observations: (1) *Ngly1$^{-/-}$* MEFs show BMP4$^{HA-Myc}$ accumulation in the ER, impaired BMP4 secretion and signaling, and induction of several ER stress markers (*Figure 4*); (2) In *Ngly1$^{-/-}$* MEFs, BMP4$^{HA-Myc}$ interacts with OS9 (*Figure 7*), an ER lectin which specifically binds misfolded glycoproteins to promote their transport to the retrotranslocation machinery (*Christianson et al., 2008*); (3) Mutations that abolish NGLY1's VCP binding impair its ER recruitment and prevent NGLY1 from deglycosylating BMP4, rescuing BMP4 signaling in *Ngly1$^{-/-}$* MEFs, and rescuing Dpp signaling in *Pngl$^{-/-}$* embryos (*Figures 5* and *6*); and (4) Pharmacological inhibition of VCP in *Ngly1$^{+/+}$* cells results in the accumulation of a glycosylated form of BMP4$^{HA-Myc}$ and induction of ER stress (*Figure 7*). We note that blocking the function of proteasome does not impair the ability of control MEFs to send the BMP4 signal, even though a deglycosylated form of BMP4 accumulates in these cells. This suggests that even though NGLY1 ultimately promotes the proteasomal degradation of misfolded BMP4 molecules, the critical NGLY1-dependent step in BMP4 signaling is retrotranslocation of misfolded BMP4 molecules from the ER.

Previous studies have reported that mutations affecting the cleavage or folding of some BMP and TGFβ family ligands can block the secretion and activity of wild-type BMP and/or TGFβ ligands in a dominant-negative fashion, likely through generation of non-productive dimers with wild-type ligands (*Lopez et al., 1992*; *Hawley et al., 1995*; *Suzuki et al., 1997b*; *Thomas et al., 1997*). Accordingly, a likely scenario for the role of NGLY1 in BMP4 signaling is that it helps clear the misfolded BMP4 precursor monomers from the ER, thereby reducing the possibility that nonfunctional dimers consisting of misfolded and properly folded BMP4 will form. This, in turn, promotes the formation of properly folded BMP4 precursor dimers that can be transported out of the ER to be cleaved to generate the functional ligand. Notably, when *N*-glycosylation sites of Dpp are ablated by NQ mutations, it no longer depends on Pngl for signaling in the fly midgut (*Figure 1*). These observations strongly suggest that the critical role played by NGLY1 in this context is direct removal of *N*-glycans from misfolded Dpp/BMP4. Considering the above-mentioned model, our data suggest that the misfolded versions of Dpp$^{3NQ}$ and BMP4$^{HA-Myc-4NQ}$ can be efficiently retrotranslocated independently of Pngl/NGLY1, potentially by a previously reported, glycosylation-independent ERAD pathway, which heavily depends on GRP78/BiP (*Ushioda et al., 2013*).

Recent studies have identified NFE2L1 as an evolutionarily-conserved, direct target of NGLY1 (*Lehrbach and Ruvkun, 2016*; *Tomlin et al., 2017*; *Lehrbach et al., 2019*). Our data identify Dpp/BMP4 as another biologically relevant target of NGLY1. Intriguingly, we find several differences between the ways NGLY1 regulates SKN-1/NFE2L1 versus Dpp/BMP4. First, deglycosylation of NFE2L1 by NGLY1 leads to its activation. However, deglycosylation of BMP4 by NGLY1 promotes its proteasomal degradation. Second, the cytoplasmic pool of NGLY1 is capable of deglycosylating

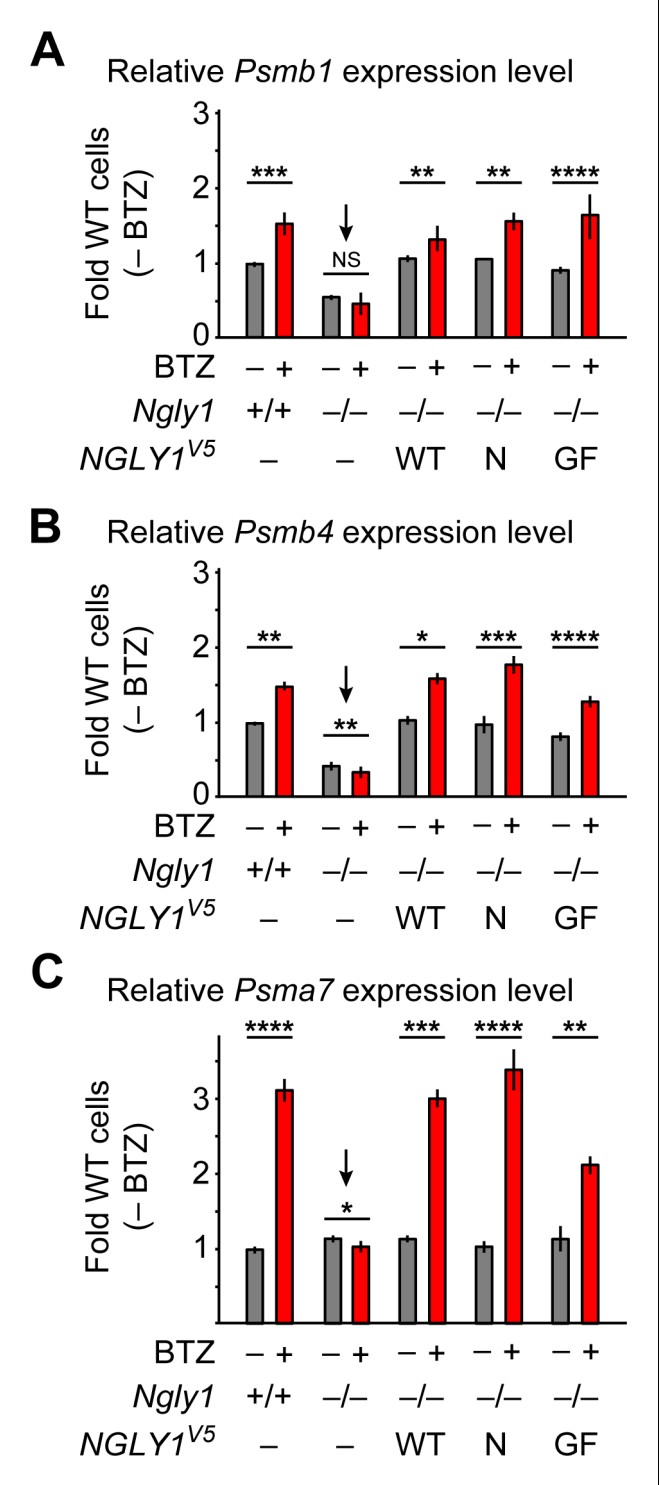

**Figure 8.** ER-associated NGLY1 is not required for NFE2L1-mediated proteasome bounce-back. (**A–C**) qRT-PCR assays for proteasome subunit genes in WT and *Ngly1* mutant MEFs with or without bortezomib (BTZ) treatment. *Ngly1* mutant MEFs were transfected with NGLY1$^{V5-WT}$ and NGLY1$^{V5}$ with mutations in VCP-binding sites (N41P and G79A/F80A). n = 3 biologically independent samples. Two-way ANOVA test with Tukey's multiple comparisons tests was used for statistical analysis. Mean ± s.e.m. is shown. *p<0.05; **p<0.01; ***p<0.001; ****p<0.0001; NS, not significant. Despite impaired ER recruitment, the mutant NGLY1 proteins retain the ability to promote the proteasome bounce-back response, which depends on NFE2L1 deglycosylation by NGLY1 (**Tomlin et al., 2017**).

*Figure 8 continued on next page*

*Figure 8 continued*

The online version of this article includes the following source data for figure 8:

**Source data 1.** Raw data and statistical analysis for panels A-C.

NFE2L1, suggesting that NFE2L1 retrotranslocation does not depend on NGLY1. However, only the ER-recruited pool of NGLY1 seems to be able to deglycosylate BMP4, suggesting a role in BMP4 retrotranslocation. Third, the key role of PNG-1/NGLY1 on SKN-1/NFE2L1 in *C. elegans* is not the removal of the *N*-glycans per se, but is the N-to-D sequence editing of NFE2L1 *N*-glycosylation sites (*Lehrbach et al., 2019*). However, our data suggest that Pngl/NGLY1 regulates BMP signaling by removing *N*-glycans from misfolded Dpp/BMP4, and that N-to-D editing is not essential for the function of Dpp/BMP4.

NGLY1 deficiency affects multiple organ systems in human patients (*Need et al., 2012*; *Enns et al., 2014*; *Lam et al., 2017*). However, despite the recent advances in understanding the function of NGLY1 and its homologs, the molecular basis for the developmental phenotypes of this disease has remained elusive. In agreement with the tissue-specific defects observed in *Pngl*-mutant flies (*Galeone et al., 2017*), pSMAD1/5 staining in E15.5 *Ngly1*$^{-/-}$ mouse embryos shows impaired BMP signaling in certain tissues, including the 4th ventricle choroid plexus. Future work will clarify whether any of the patient phenotypes (including neurological deficits and orthopedic features [*Cahan and Frick, 2019*]) are caused by altered BMP signaling.

## Materials and methods

### *Drosophila* strains, genetics and eclosion tests

Flies were grown and crossed on standard food containing cornmeal, molasses, and yeast at room temperature (~22°C). The following fly strains were used in this study: *Mef2-GAL4* (Bloomington *Drosophila* Stock Center (BDSC) #27390), *UAS-CD8::GFP*, *UAS-lacZ*, *UAS-dpp-GFP* (BDSC #53716) (*Teleman and Cohen, 2000*), *dpp-GAL4* (BDSC #1553), *Mi{MIC}dpp*$^{MI03752}$ (BDSC #36399) *PBac{RB} e00178* (Exelixis at Harvard Medical School), *Pngl*$^{ex14}$ (*Funakoshi et al., 2010*), *UAS-Pngl*$^{RNAi}$ KK101641 (Vienna *Drosophila* Resource Center), *UAS-attB-NGLY1*$^{WT}$*-VK31* (*Galeone et al., 2017*), *UAS-attB-NGLY1*$^{N41P}$*-VK31*, *UAS-attB-NGLY1*$^{F70A/G80A}$*-VK31*, *dpp*$^{HA}$ and *dpp*$^{HA-3NQ}$ (this study). For survival (eclosion test), the expected ratio of offspring was calculated based on Mendelian inheritance for each genotypic class and the observed/expected ratio is reported as a percentage.

### Generation of *dpp*$^{HA}$ knock-in alleles

The details for the generation of these alleles will be reported elsewhere. In brief, utilizing the *attP* sites in a MiMIC transposon inserted in the *dpp* locus (*Mi{MIC}dpp*$^{MI03752}$), about 4.4 kb of the *dpp* genomic sequences containing the second (last) coding exon of *dpp* and its flanking sequences was inserted in the intron between *dpp*'s two coding exons. To generate the *dpp*$^{HA}$ allele, an HA tag was inserted in the active domain at the same position as the *UAS-dpp-GFP* strain used in our study (*Teleman and Cohen, 2000*). To generate the *dpp*$^{HA-3NQ}$ allele, in addition to the HA tag, the three *N*-glycosylation sites in this exon were mutated to Q. In both cases, an *FRT* site and a *3XP3-mCherry* element were inserted after the exon. Insertions with proper orientation were selected by PCR. Then, utilizing the inserted *FRT* site and the *FRT* site in the downstream *PBac{RB}e00178* transposon, the *3XP3-mCherry* element and the endogenous last exon of *dpp* downstream of the newly inserted exon were removed by heat shock-mediated expression of Flippase (FLP) in *hsFLP; PBac{RB}e00178/ dpp*$^{HA}$*-FRT-3XP3-mCherry* and *hsFLP; PBac{RB}e00178/dpp*$^{HA-3NQ}$*-FRT-3XP3-mCherry* animals. The resulting knock-in alleles would acquire the mini-white$^+$ (*mini-w*$^+$) from *PBac{RB}e00178* and contain HA-tagged *dpp* exon (*Figure 1B*, wild-type and 3NQ).

### Generation of *NGLY1* overexpression *Drosophila* transgenes

Human *NGLY1* cDNA in *pCMV6-AC* vector (clone SC320763, OriGene) was used as a template for site-directed mutagenesis to introduce the c.121-122AA > CC and c.236-241GCCTTT > GCCGCT mutations, which result in the generation of *NGLY1*$^{N41P}$ and *NGLY1*$^{G79A/F80A}$, respectively. The following primers were used (5' to 3'): hNG1-N41P-for CTCACCTATGCTGACCCCATCC

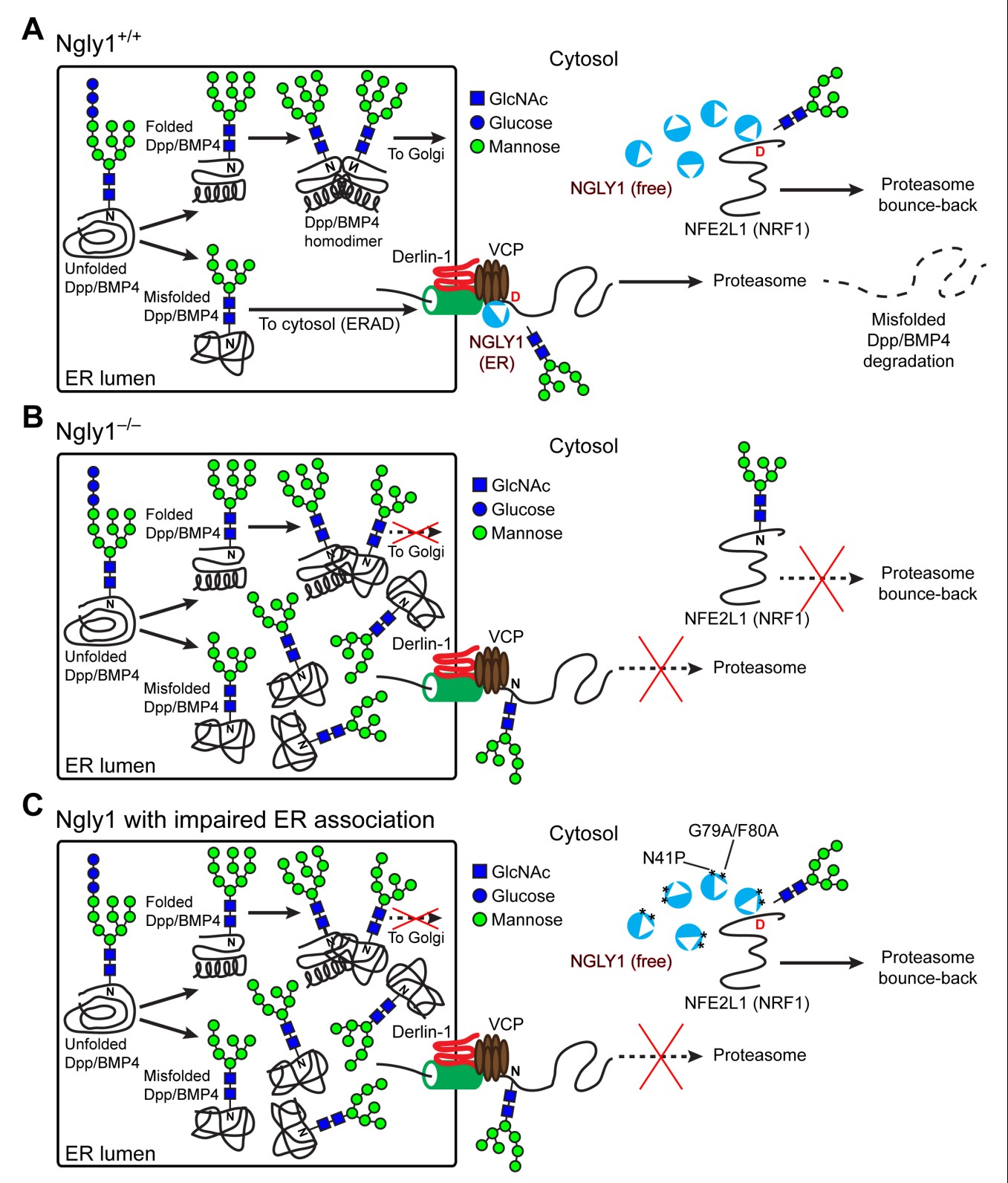

**Figure 9.** Cytosolic NGLY1 is dynamically recruited to retrotranslocon machinery for misfolded BMP4 retrotranslocation. (**A**) Proposed model: Upon ER stress due to the generation of misfolded BMP4/Dpp in the ER, cytosolic NGLY1 is recruited to the ER in a VCP-dependent manner and deglycosylates BMP4 molecules to promote their retrotranslocation and proteasomal degradation. Deglycosylation and N-to-D sequence editing of NFE2L1 by NGLY1 allows the proteasome bounce-back response. (**B**) Loss of *Ngly1* leads to impaired BMP4 deglycosylation and retrotranslocation and thereby results in

*Figure 9 continued on next page*

*Figure 9 continued*

the accumulation of misfolded BMP4 molecules in the ER and a severe decrease in BMP4 signaling. It also impairs NFE2L1 deglycosylation/sequence editing, which leads to the loss of the proteasome bounce-back response. (C) The cytosolic NGLY1 pool is sufficient for NFE2L1 deglycosylation/ sequence editing and proteasome bounce-back but not for BMP4 deglycosylation and retrotranslocation. Asterisks mark NGLY1 mutations which abolish VCP binding and ER recruitment.

TCAGAAACCC hNG1-N41P-rev GGGTTTCTGAGGATGGGGGTCAGCATAGGTGAG hNG1-G79F80-for GTTTATTTGAAATGGCCGCTGAAGAGGGAGAAAC hNG1-G79F80-rev GTTTCTCCCTC TTCAGCGGCCATTTCAAATAAAC cDNAs of mutant versions of *NGLY1* were transferred from *pCMV6-AC* to *pUAST-attB* vector by *EcoRI-XhoI* double digestion and ligation, exactly as performed before for wild-type *NGLY1* construct (*Galeone et al., 2017*), verified by sequencing, and integrated into the same docking site (*VK31*) by ΦC31-mediated transgenesis (*Venken et al., 2006*; *Bischof et al., 2007*). Embryo injections were performed by GenetiVision (Houston, USA).

## *Drosophila* immunostaining, western blotting, and PNGase F digestion

Embryos were fixed by formaldehyde-based fixation. Wing imaginal discs were dissected and fixed from late third instar larvae. Antibodies were rat anti-HA 1:250 (clone 3F10, Roche), rabbit anti-pSMAD3 1:250 (ab52903, Abcam), rabbit anti-NGLY1 (HPA036825, Sigma-Aldrich) 1:200, goat anti-rat-Cy3 1:500, goat anti-rabbit-Cy5 1:500 (Jackson ImmunoResearch Laboratories). Confocal images were taken with a Leica TCS-SP8 microscope. All images were acquired using Leica LAS-SP software. Amira 5.2.2 and Adobe Photoshop CS6 were used for processing and Figures were assembled in Adobe Illustrator CS6. Proteins were extracted from whole embryo in lysis buffer containing protease inhibitor cocktail (Promega). PNGase F treatment was performed following New England BioLabs' protocol. Specifically, 20 µg protein was dissolved in denaturing buffer containing 0.5% SDS + 40 mM DTT (provided with the P0704S PNGase F) for 10 min at 100˚C. After addition of Glycobuffer two sample was treated with PNGase (15,000 U) and incubated at 37˚C for 1 hr. The following antibodies were used: mouse anti-GFP 1:1000 (Sigma Aldrich) and goat anti-mouse-HRP 1:2000 (Jackson ImmunoResearch Laboratories). Western blots were developed using Pierce ECL Western Blotting Substrates (Thermo Scientific). The bands were detected using a ChemiDoc MP system (Bio-Rad). At least three independent immunoblots were performed for each experiment.

## Immunohistochemistry and immunostaining of mouse embryonic tissues

The mouse strain used in this study is *Ngly1*[em4Lutzy], a functional null allele of *Ngly1* available from The Jackson Laboratory (Stock #027060). The allele was created by CRISPR/Cas9 technology followed by non-homologous end joining, which resulted in an 11 bp deletion in exon 8 (Supporting *Figure 2*). The strain is kept on a C57BL/6J background. The mice were maintained in the pathogen-free barrier facilities at Jackson Laboratory (Bar Harbor, ME) and at Baylor College of Medicine (Houston, TX), and the studies were conducted in accordance with approved institutional animal care and use committee (IACUC) protocols 99066 (Jackson Laboratory) and AN-6012 (Baylor College of Medicine). The animals had full-time access to food and water and were maintained under 12 hr light/12 hr dark cycle. Embryos were dissected in ice-cold 1x PBS placing them in labeled, appropriately sized glass vials. Embryos were fixed in 4% PFA at 4˚C overnight. Fixed embryos were washed with 1x PBS and were cryoprotected through a series of sucrose solutions (5%, 10%, 20%). Embryos were cleared from sucrose solution and embedded in the Optimal Cutting Temperature (OCT) compound. Samples were sectioned at 5 µm and stained with hematoxylin and eosin (H&E staining) using the standard protocol. For immunostaining, sections underwent heat-induced epitope retrieval and probed with rabbit anti-pSMAD1/5 1:500 (Cell Signaling, 41D10), followed by secondary antibody staining with goat anti-rabbit Cy3 (Jackson ImmunoResearch, 111-165-144). The stained sections were mounted in Vectashield DAPI (Vector Laboratories) and imaged with a Zeiss 710 confocal microscope.

# Construction of *pIRES-NGLY1*[WT-V5-His6]*-DsRED*, *pIRES-NGLY1*[N41P-V5-His6]*-DsRED* and *pIRES-NGLY1*[G79A/F80A-V5-His6]*-DsRED* expression vectors

For construction of *pIRES-NGLY1*[WT-V5-His6]*-DsRED*, cDNA sequence of *NGLY1* was amplified by PCR using *pCMV6-huPNGase-WT* as a template with the following primer set (5′ to 3′):

> pIRES-EcoRI-huNGLY1-F CTCAAGCTTCGAATTCTCAAGCATGGCGGCGGCG
> v5His6-tga-Sall-pIRES-R    CCGCGGTACCGTCGACTCAATGGTGATGGTGGTGATGCGTAGAATCGAGACCGAG

The resultant NGLY1[V5-His6] fragment was cloned into *pIRES-DsRed2-Express2*, which was digested with *EcoR*I and *Sal*I, using In-Fusion HD cloning kit (TaKaRa, Kyoto, Japan) according to the manufacturer's instructions. Construction of *pIRES-NGLY1*[N41P-V5-His6]*-DsRED* and *pIRES-NGLY1*[G79A/F80A-V5-His6]*-DsRED* was carried out using QuickChange Lightning Site-Directed Mutagenesis Kit (Agilent Technologies, Inc, Tokyo, Japan) according to the manufacturer's instructions with *pIRES-DsRed2-Express2-NGLY1*[V5-His6] as a template and the following primer sets (5′ to 3′):

> NGLY1-N41P-F ATTAGGGTTTCTGAGGATGGGGTCAGCATAGGTGAGCAGC
> NGLY1-N41P-R GCTGCTCACCTATGCTGACCCCATCCTCAGAAACCCTAAT
> NGLY1-G79A_F80A-F    GATGAGATGTGTTTCTCCCTCTTCAGCGGCCATTTCAAATAAACATTCAACAGC
> NGLY1-G79A_F80A-R    GCTGTTGAATGTTTATTTGAAATGGCCGCTGAAGAGGGAGAAACACATCTCATC

## Construction of expression vectors of NGLY1 mutants for cell free system

Fragments of *NGLY1*[V5-His6], *NGLY1*[N41P-V5-His6] and *NGLY1*[G79A/F80A-V5-His6] were released from the corresponding *pIRES-DsRed2-Express2* constructs by digestion with *Xho*I and *BamH*I. The purified fragments were ligated into equivalent sites of *pEU-E01-MCS*, an expression vector for the cell free system (CellFree Science, Ehime, Japan).

## Preparation of fluorescence-labeled substrates, BODIPY-asialoglycopeptide (ASGP)

Detailed method for preparation of fluorescently labeled glycopeptides will be published elsewhere. Briefly, N-terminal labeling of ASGPs (NH2-KVAN(CHO)KT-COOH) (FUSHIMI Pharmaceutical Co. Ltd, Tokushima, Japan) by BODIPY-N-Hydroxysuccinimide (BODIPY-NHS) was carried out according to the manufacturer's instructions (Thermo Scientific, Waltham, MA), and the labeled peptides were purified using reversed-phase HPLC.

## Protein expression by cell free system

Protein expression was carried out using ENDEXTTechnology, Protein Research Kit (H) (Cell Free Science, Ehime, Japan), according to the manufacture's protocols. Briefly, for transcription of wild-type and mutant *NGLY1*, a mixture of 2 µg of the expression plasmid and transcription buffer (LM premix) was incubated for 1 hr at 37°C. Then, the transcribed mRNA and wheat germ extract were mixed. The mixture was transferred to the bottom of translation buffer to form a bi-layer with the mixture of wheat germ extract and mRNA (lower layer) and the translation buffer (upper layer). The bilayer solution was incubated for 24 hr at 15°C for translation reaction.

## Purification of NGLY1[V5-His6] from cell-free system

For affinity purification of NGLY1[V5-His6], 15 µl of Ni-Sepharose resin (GE Healthcare Life Sciences, Piscataway, NJ) was added to the translated reaction mixture and incubated for 1 hr at 4°C with rotation. After washing the NGLY1[V5-His6]-bound Ni-Sepharose resin with 400 µl of washing buffer (20 mM Na-phosphate pH 7.5, 300 mM NaCl, and 20 mM Imidazole), NGLY1[V5-His6] proteins were eluted by elution buffer (20 mM Na-phosphate pH 7.5, 300 mM NaCl, and 500 mM Imidazole). Concentration of the protein and removal of imidazole were carried out using Amicon Ultra-0.5 mL centrifugal filter (30 k MWCO) (Millipore, Bedford, MA).

## Activity assay of purified NGLY1 from germ wheat extract

An aliquot of 10 µl of purified NGLY1$^{V5-His6}$ was incubated with 50 pmol of BODIPY-ASGP in 40 µl of the following NGLY1 reaction buffer for 10 min at 37°C: 10 mM Tris-HCl pH 7.5, 50 mM sucrose, 1 mM DTT, 1 mM Pefabloc SC (Roche Diagnostics, Tokyo, Japan), and cOmplete protease inhibitor cocktail (Roche Diagnostics). Reaction was terminated by adding 50 µl of 100% EtOH and centrifugation at 15,000 rpm for 5 min. The resultant supernatant was collected and evaporated to dryness in a Speed-Vac concentrator. Enzyme activities were normalized by NGLY1 protein concentrations measured by BCA method (Thermo Scientific), according to manufacturer's instructions, using bovine serum albumin as a standard.

## Separation and quantification of the reaction product by HPLC

Substrate and reaction product were separated by HPLC using InertSustain C18 HP (3 µm, 3.0 × 150 mm, GL Science). The elution condition was as follows: eluent A, distilled water containing 0.1% Trifluoroacetic Acid (TFA); eluent B, 100% acetonitrile containing 0.1% TFA. The column was equilibrated with eluent A/eluent B (60/40) at a flow rate of 0.45 ml/min. After injecting a sample, the concentration of eluent B was increased linearly from 40% to 95% over 25 min. BODIPY-ASGP was detected by measuring fluorescence (λ excitation 503 nm, λ emission 512 nm).

## Generation of the *Bmp4$^{HA-Myc-4NQ}$* expression plasmid

The *pCS2+Bmp4$^{HA-Myc}$* vector is a gift from Jan Christian. The HA and Myc tags are inserted in the following positions in the mouse BMP4 protein (NP_031580): $^{59}$ATLypydvpdyaLQM$^{64}$ (HA tag in lowercase, in the pro-domain) and $^{294}$PKHeqkliseedlHPQ$^{299}$ (Myc tag in lowercase, in the active domain). Numbers indicate amino acid positions in the untagged mouse BMP4 protein. The *Bmp4$^{HA-Myc}$* open reading frame (ORF) in this vector was used as a template for synthesizing double-stranded DNA that harbors asparagine (N) to glutamine (Q) mutations in all four *N*-glycosylation sites of BMP4 (*Bmp4$^{HA-Myc-4NQ}$*). Gene synthesis was performed by GENEWIZ (South Plainfield, NJ). The mutated ORF was transferred to *pCS2+* by *EcoRI-XhoI* double digestion and ligation and verified by Sanger sequencing. *pCMV-GFP* was a gift from Connie Cepko (Addgene plasmid #11153; http://n2t.net/addgene:11153; RRID:Addgene_11153) (*Matsuda and Cepko, 2004*).

## MEF cell culture and transfection, 3T3-L1 cell culture, and treatment with BMP4 and chemical inhibitors of the proteasome, NGLY1 and VCP

Wild-type and mutant NGLY1 MEFs were described previously (*Huang et al., 2015*) and confirmed by western blotting and immunostaining for NGLY1. MEFs and 3T3-L1 cells (ATCC, CL-173, confirmed by differentiation assays) were cultured in Dulbecco's modified Eagle's medium (Sigma-Aldrich) supplemented with 10% FBS and antibiotics (100 U/mL penicillin G, 100 ng/mL streptomycin; Sigma-Aldrich) at 37°C in humidified air containing 5% (vol/vol) $CO_2$. All three cell lines were found to be free from mycoplasma. MEFs were transfected using FuGENE HD transfection reagent (Roche Applied Sciences) according to the manufacturer's instructions. Briefly, 5 × 10$^5$ cells were seeded 12–16 hr (h) before transfection and 1 µg plasmid/3 µL transfection reagent was mixed in 150 µL opti-MEM (Invitrogen). For co-transfection, 1 µg plasmid/3 µL transfection reagent was used for each plasmid. The above-mentioned *pCS2+Bmp4$^{HA-Myc}$* and *pIRES-NGLY1$^{V5-His6}$-DsRED* vectors were used to express wild-type and mutant BMP4 and NGLY1 proteins in MEFs. All the transiently transfected cells were incubated for 48 hr before harvesting for analysis. To induce protein misfolding in the ER, cells were treated for 30 min with 2 mM of DTT (Sigma Aldrich). DTT was dissolved in distilled $H_2O$ in a 1 M stock and added freshly to cells with a final DTT concentration of 0.2% (vol/vol). To inhibit VCP, cells were treated for 6 hr at 5 µM of NMS-873 (Sigma Aldrich). To block proteasome, cells were treated for 6 hr at 10 nM of BTZ (Cayman Chemical). Master solutions of NMS-873 and BTZ were made in DMSO at 50 mM and 10 mM, respectively, and were used freshly or stored at −80°C without repeated freeze-thaw cycles. NMS-873 and BTZ were added to the cells with a final concentration of DMSO of 0.1% (vol/vol) in each well. To inhibit the function of NGLY1, 3T3-L1 cells were treated with Z-VAD-fmk (R and D, #2163) for 24 hr at 20 µM and 50 µM. To inhibit caspases without affecting NGLY1 function, 3T3-L1 cells were treated with Q-VD-OPh (Cayman Chemical, #15260) for 24 hr at 20 µM. The cells were allowed to recover for 1 hr before protein extraction, as described before (*Tomlin et al., 2017*). To assess the function of MEFs in responding to BMP4,

recombinant BMP4 (R and D Systems, catalog # 314 BP) was added to the culture medium at 0.5 nM for 2 hr, followed by western blotting for pSMAD1/5.

## Western blot analysis and PNGase F/Endo H digestion on MEFs

Cells were lysed in cold lysis buffer (20 mM Tris-HCl, pH 7.4, 150 mM NaCl and 1% Triton X-100) in the presence of protease inhibitors (Sigma-Aldrich) for 30 min (min) on ice. Cell media were collected, and proteins were precipitated by TCA/Acetone general protocol. Total protein concentrations were determined by BCA assay (Pierce). For immunoprecipitation experiments, cells were scraped off Petri dishes in cold PBS, centrifuged at 400 g for 5 min and the pellet was collected. To detect the interaction between OS9 and the HA-tagged BMP4, immunoprecipitation was performed using HA-tag IP/Co-IP kit (PIERCE). Briefly, cells were lysed in NP40 lysis buffer (50 mM Tris-HCl, pH 7.5, 150 mM NaCl, 1% NP40 [vol/vol], 10% glycerol) with protease inhibitors (Roche). Cell lysates were incubated with anti-HA-agarose (Profound HA, Pierce) overnight at 4°C and the precipitated proteins eluted according to the manufacturer's protocol. Protein samples were separated on 7.5% Bis-Tris precast gels (Bio-Rad) and transferred to nitrocellulose membranes using the turbo transfer system (Bio-Rad) according to manufacturer's instructions. The following antibodies were used: mouse anti-HA 1:1000 (Sigma-Aldrich, GT4810), rabbit anti-pSMAD 1:500 (Cell Signaling, 41D10), mouse anti-Myc 1:500 (Cell Signaling, 9B11), mouse anti-FK1 1:500 (Sigma-Aldrich, 04–262), mouse anti-KDEL 1:500 (Santa Cruz Biotechnology, 10C3), rabbit anti-VCP 1:500 (Santa Cruz Biotechnology, sc-20799 [*Radhakrishnan et al., 2014*]), rabbit anti-IRE1α 1:1000 (Cell Signaling, 14C10), rabbit anti-pIRE1α 1:1000 (Novus Biologicals, 100–2323), rabbit anti-OS9 1:1000 (Abcam, ab109510), and mouse anti-V5 1:1000 (FUJIFILM Wako Pure Chemical Corporation, 6F5). Primary and horseradish peroxidase-conjugated secondary antibodies were diluted in 5% milk in TBST. Detection was carried out with ECL western Blotting detection reagent (Biorad). Images were detected with ChemiDoc MP (Biorad) and quantified by Fiji analysis software. PNGase F and Endo H treatments were performed following New England BioLabs' protocol. Specifically, 20 μg protein or 20 μg of precipitated protein (resuspended in lysis buffer) were dissolved in denaturing buffer containing 0.5% SDS + 40 mM DTT (provided with the P0704S PNGase F and P0703L Endo H kits) for 10 min at 100°C. After addition of Glycobuffer three samples were treated with PNGase and Endo H enzymes (15,000 U) and incubated at 37°C for 1 hr. Lysates of control samples were prepared the same way but not treated with deglycosylating enzymes.

## Confocal microscopy on MEFs

Transfected cells were grown on glass coverslips for 24 hr, washed with PBS and fixed with 4% paraformaldehyde (PFA; Sigma- Aldrich) for 15 min. After quenching PFA with 50 mM NH4Cl for 15 min, cells were washed with PBS and permeabilized in blocking buffer (0.1% saponin/10% FBS in PBS) for 1 hr. Coverslips were then incubated overnight with appropriate primary antibodies and for 1 hr with Alexa-Fluor-conjugated secondary antibodies. Coverslips were mounted on glass slides with Vectashield DAPI (Vector Laboratories). Images were taken with a Nikon AR1 confocal microscope using a 63°ø immersion objective (Nikon). For quantitative colocalization analysis, five independent pictures were acquired and analyzed with the Coloc2 plugin in Fiji software.

## Real-time qRT-PCR

RNA samples were obtained using the RNeasy kit (Qiagen) according to the manufacturer's instructions. RNA was quantified using the Nano-Drop 8000 (Thermo Fisher). cDNA was synthesized using the SuperScript VILO cDNA Synthesis Kit (Thermo Fisher). Real-time quantitative RT–PCR on cDNAs was carried out with SsoFast EvaGreen Supermix using CFX96 Real-Time PCR detection system (Bio-Rad) with the following conditions: 95°C for 5 min; (95°C for 10 s; 60°C for 10 s; 72°C for 15 s)×40 cycles. For expression profile, the qRT-PCR results were normalized to an internal control (GAPDH). The following oligonucleotide sequences were used to assess proteasomal gene expression, as described previously (5' to 3') (*Yang et al., 2018*):

    m-Psmb1-F CCTTCAACGGAGGTACTGTATTG
    m-Psmb1-R GGGCTATCTCGGGTATGAATTG
    m-Psmb4-F CGAGTCAACGACAGCACTAT
    m-Psmb4-R ATCTCCCAACAGCTCTTCATC

m-Psma7-F CGAGTCTGAAGCAGCGTTAT
m-Psma7-R AGTCTGATAGAGTCTGGGAGTG

## Statistical analysis

GraphPad Prism was used for statistical analysis. The statistical tests performed for each experiment are indicated in figure legends. Numerical data are shown as mean ± SEM or mean ± SD, as indicated. p Values of less than 0.05 were considered statistically significant.

## Acknowledgements

We thank Senthil Radhakrishnan for discussions; Selina Dwight, Kevin Lee and Marco Sardiello for comments on the manuscript; Giulia Vitale and Mario Lopez for excellent technical assistance; The Bloomington *Drosophila* Stock Center (NIH P40OD018537), the Developmental Studies Hybridoma Bank, Jan Christian, Senthil Radhakrishnan and Sandhya Thomas for reagents. This work was supported by grants from the Grace Science Foundation (to AZ, HJN, and TS), the NIH (R35GM130317 to HJN), H2020-MSCA individual fellowship (844147) and Buzzati-Traverso fellowship (to AG), Fondazione AIRC per la Ricerca sul Cancro (20661) and Worldwide Cancer Research (18-0399) (to TV). SM was supported by JSPS Postdoctoral Fellowship for Research Abroad and the Research Fund Junior Researchers of the University of Basel, and is currently supported by an SNSF Ambizione grant (PZ00P3_180019). Imaging was performed at the Confocal Microscopy Core of the BCM IDDRC (U54HD083092; the Eunice Kennedy Shriver NICHD) and Unitech NOLIMITS microscopy facility core of University of Milan.

## Additional information

### Funding

| Funder | Grant reference number | Author |
|---|---|---|
| Grace Science Foundation | Research grant | Tadashi Suzuki<br>Aamir Zuberi<br>Hamed Jafar-Nejad |
| National Institutes of Health | R35GM130317 | Hamed Jafar-Nejad |
| H2020 Marie Skłodowska-Curie Actions | H2020-MSCA individual fellowship #844147 | Antonio Galeone |
| Private Foundation in Italy | Buzzati-Traverso fellowship | Antonio Galeone |
| Fondazione AIRC per la Ricerca sul Cancro | grant # 20661 | Thomas Vaccari |
| Worldwide Cancer Research | grant #18-0399 | Thomas Vaccari |
| Japan Society for the Promotion of Science | JSPS Postdoctoral Fellowship for Research Abroad | Shinya Matsuda |
| University of Basel | Research Fund Junior Researchers of the University of Basel | Shinya Matsuda |
| SNSF Ambizione | PZ00P3_180019 | Shinya Matsuda |

The funders had no role in study design, data collection and interpretation, or the decision to submit the work for publication.

### Author contributions

Antonio Galeone, Conceptualization, Formal analysis, Funding acquisition, Investigation, Writing - original draft, Writing - review and editing; Joshua M Adams, Maximiliano F Presa, Ashutosh Pandey, Seung Yeop Han, Formal analysis, Investigation, Writing - review and editing; Shinya Matsuda, Yuriko Tachida, Hiroto Hirayama, Resources, Formal analysis, Investigation, Writing - review and editing; Thomas Vaccari, Tadashi Suzuki, Aamir Zuberi, Resources, Funding acquisition, Writing - review and

editing; Cathleen M Lutz, Markus Affolter, Resources, Writing - review and editing; Hamed Jafar-Nejad, Conceptualization, Supervision, Funding acquisition, Investigation, Writing - original draft, Project administration, Writing - review and editing

### Author ORCIDs
Shinya Matsuda (iD) http://orcid.org/0000-0002-7541-7914
Markus Affolter (iD) http://orcid.org/0000-0002-5171-0016
Hamed Jafar-Nejad (iD) https://orcid.org/0000-0001-6403-3379

### Ethics
Animal experimentation: This study was performed in strict accordance with the recommendations in the Guide for the Care and Use of Laboratory Animals of the National Institutes of Health. The mice were maintained in the pathogen-free barrier facilities at Jackson Laboratory (Bar Harbor, ME) and at Baylor College of Medicine (Houston, TX). The studies were conducted in accordance with approved institutional animal care and use committee (IACUC) protocols 99066 (Jackson Laboratory) and AN-6012 (Baylor College of Medicine).

### Decision letter and Author response
Decision letter https://doi.org/10.7554/eLife.55596.sa1
Author response https://doi.org/10.7554/eLife.55596.sa2

## Additional files
### Supplementary files
• Transparent reporting form

### Data availability
All data generated or analysed during this study are included in the manuscript and supporting files.

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

## Appendix 1

<div style="border-left: 6px solid #1a7fc0; padding-left: 1em;">

# Key Resources Table

**Appendix 1—key resources table**

| Reagent type (species) or resource | Designation | Source or reference | Identifiers | Additional information |
|---|---|---|---|---|
| Gene (*D. melanogaster*) | *Pngl* | GenBank | FLYB: FBgn0033050 | |
| Gene (*Mus musculus*) | *Ngly1* | GenBank | Gene ID: 59007 | |
| Gene (*Homo sapiens*) | *NGLY1* | GenBank | Gene ID: 55768 | |
| Strain, strain background (*Escherichia coli*) | TOP10. Genotype: F- mcrA Δ(mrr-hsdRMS-mcrBC) φ80lacZΔM15 ΔlacX74 recA1 araD139 Δ(araleu)7697 galU galK rpsL (StrR) endA1 nupG | Life Technologies | C404010 | Chemically Competent Cells |
| Genetic reagent (*D. melanogaster*) | *Mef2-GAL4* | Bloomington *Drosophila* Stock Center | BDSC: 27390; FLYB: FBti0115746 | FlyBase symbol: *P{GAL4-Mef2.R}3* |
| Genetic reagent (*D. melanogaster*) | *UAS-dpp-GFP* | Bloomington *Drosophila* Stock Center | BDSC: 53716; FLYB: FBti0157003 | FlyBase symbol: *P{UAS-dpp.GFP.T}3* |
| Genetic reagent (*D. melanogaster*) | *dpp-GAL4* | Bloomington *Drosophila* Stock Center | BDSC: 1553; FLYB: FBti0002123 | FlyBase symbol: *P{GAL4-dpp.blk1}40C.6* |
| Genetic reagent (*D. melanogaster*) | *Mi{[MIC]}dpp$^{MI03752}$* | Bloomington *Drosophila* Stock Center | BDSC: 36399; FLYB: FBti0145223 | FlyBase symbol: *Mi{MIC}dpp$^{MI03752}$* |
| Genetic reagent (*D. melanogaster*) | *PBac{[RB]}* | Exelixis at Harvard Medical School | FLYB:e00178 FBti0046265 | FlyBase symbol: *PBac{RB}e00178* |
| Genetic reagent (*D. melanogaster*) | *Pngl$^{ex14}$* | **Funakoshi et al., 2010** | FLYB: FBal0244826 | FlyBase symbol: *Pngl$^{ex14}$* |
| Genetic reagent (*D. melanogaster*) | *UAS-Pngl$^{RNAiKK101641}$* | Vienna *Drosophila* Resource Center | VDRC: v103607 FLYB: FBst0475465 | FlyBase symbol: *P{KK101641}VIE-260B* |
| Genetic reagent (*D. melanogaster*) | *UAS-attB-NGLY1$^{WT-VK31}$* | **Galeone et al., 2017** | | Fly strain carrying human *UAS-cDNA* of wild-type *NGLY1* |

</div>

*Appendix 1—key resources table continued*

| Reagent type (species) or resource | Designation | Source or reference | Identifiers | Additional information |
|---|---|---|---|---|
| Genetic reagent (*D. melanogaster*) | *UAS-attB-NGLY1*$^{N41P}$-*VK31* | This paper | | Fly strain carrying human *UAS-cDNA* of *NGLY1* with mutation in VCP binding site |
| Genetic reagent (*D. melanogaster*) | *UAS-attB-NGLY1*$^{F70A/G80A}$-*VK31* | This paper | | Fly strain carrying human *UAS-cDNA* of *NGLY1* with mutation in VCP binding site |
| Genetic reagent (*D. melanogaster*) | *dpp*$^{HA}$ | This paper | | Fly strain carrying endogenous *HA-tag* of *dpp* active domain |
| Genetic reagent (*D. melanogaster*) | *dpp*$^{HA-3NQ}$ | This paper | | Fly strain carrying endogenous *HA-tag* of *dpp* active domain and N to Q mutations in *N*-glycosylation sites |
| Strain, strain background *Mus musculus* | *Ngly1*$^{em4Lutzy}$ kept on a C57BL/6J background | The Jackson Laboratory | Stock #027060 | Mouse strain carrying an 11 bp deletion in exon 8 of the mouse *Ngly1* |
| Cell line (*Mus musculus*) | Mouse Embryonic Fibroblast (MEF) cells | **Huang et al., 2015** | | Established from C57BL/6J background |
| Cell line (*Mus musculus*) | 3T3-L1 cells | American Type Culture Collection | ATCC, CL-173 | From Dr. Sandhya Thomas |
| Recombinant DNA reagent | Human *NGLY1* cDNA in pCMV6-AC (plasmid) | OriGene | clone SC320763 | Human untagged cloning vector |
| Recombinant DNA reagent | *pIRES-DsRed2-Express2* | Clontech | Cat# 632540 | Expressing vector backbone construct |
| Transfected construct (*Homo sapiens*) | *pIRES-NGLY1*$^{WT-V5-His6}$-*DsRED* | This paper | | transfected *NGLY1* construct (human) |
| Transfected construct (*Homo sapiens*) | *pIRES-NGLY1*$^{N41P-V5-His6}$-*DsRED* | This paper | | transfected *NGLY1* construct (human) |
| Transfected construct (*Homo sapiens*) | *pIRES-NGLY1*$^{G79A/F80A-V5-His6}$-*DsRED* | This paper | | transfected *NGLY1* construct (human) |
| Transfected construct (*Homo sapiens*) | *pCS2+Bmp4*$^{HA-Myc}$ | Gift from Dr. Jan Christian | mouse BMP4 protein (NP_031580) | Double-tagged mouse BMP4 construct |
| Transfected construct (*Homo sapiens*) | *pCS2+Bmp4*$^{HA-Myc-4NQ}$ | This paper | mouse BMP4 protein (NP_031580) | Double-tagged mouse BMP4 construct with N to Q mutations in *N*-glycosylation sites |
| Transfected construct (*Homo sapiens*) | *pUAST-attB-NGLY1*$^{WT}$ | This paper | | Fly expressing vector carrying human *UAS-cDNA* of wild-type *NGLY1* |

*Appendix 1—key resources table continued*

| Reagent type (species) or resource | Designation | Source or reference | Identifiers | Additional information |
|---|---|---|---|---|
| Transfected construct (*Homo sapiens*) | *pUAST-attB-NGLY*<sub>N41P</sub> | This paper | | Fly expressing vector carrying human *UAS-cDNA* of wild-type *NGLY1* with mutation in VCP binding site |
| Transfected construct (*Homo sapiens*) | *pUAST-attB-NGLY*<sub>F70A/G80A</sub> | This paper | | Fly expressing vector carrying human *UAS-cDNA* of wild-type *NGLY1* with mutation in VCP binding site |
| Antibody | anti-HA (Rat monoclonal) | Roche | Cat# 11867423001 RRID:AB_390918 | IF (1:250) |
| Antibody | anti- pSMAD3 (Rabbit monoclonal) | Abcam | Cat# ab52903 RRID:AB_882596 | IF (1:250) |
| Antibody | anti-NGLY1 (Rabbit polyclonal) | Sigma-Aldrich | Cat# HPA036825 RRID:AB_10672231 | IF (1:200) |
| Antibody | anti-GFP (Mouse monoclonal) | Sigma-Aldrich | Cat# G1546 RRID:AB_1079024 | WB (1:1000) |
| Antibody | anti-pSMAD1/5 (Rabbit monoclonal) | Cell Signaling | Cat# 9516 RRID:AB_491015 | IF (1:500) WB (1:500) |
| Antibody | anti-HA (Mouse monoclonal) | Sigma-Aldrich | Cat# SAB2702217 RRID:AB_2750919 | IF(1:500) WB (1:1000) |
| Antibody | anti-myc (Mouse monoclonal) | Cell Signaling | Cat# 2276 RRID:AB_331783 | WB (1:500) |
| Antibody | Anti-FK1 (Mouse monoclonal) | Sigma-Aldrich | Cat# 04–262 RRID:AB_612094 | WB (1:500) |
| Antibody | anti-KDEL (Mouse monoclonal) | Santa Cruz Biotechnology | Cat# sc-58774 RRID:AB_784161 | WB (1:500) |
| Antibody | anti-VCP (Mouse Polyclonal) | Santa Cruz Biotechnology | Cat# sc-20799 RRID:AB_793930 | WB (1:500) |
| Antibody | anti-IRE1$\alpha$ (Rabbit monoclonal) | Cell Signaling | Cat# 3294 RRID:AB_823545 | WB (1:1000) |
| Antibody | anti-pIRE1$\alpha$ (Rabbit polyclonal) | Novus Biologicals | Cat# 100–2323 RRID:AB_10145203 | WB (1:1000) |
| Antibody | anti-OS-9 (Rabbit monoclonal) | Abcam | Cat# ab109510 RRID:AB_2848681 | WB (1:1000) |
| Antibody | anti-V5 (Mouse monoclonal) | FUJIFILM Wako Pure Chemical Corporation | Cat# 4548995010711 | WB (1:1000) |
| Sequence-based reagent | hNG1-N41P-for | This paper | PCR primers | CTCACCTATGCTGACCC-CATCCTCAGAAACCC |

*Appendix 1—key resources table continued*

| Reagent type (species) or resource | Designation | Source or reference | Identifiers | Additional information |
|---|---|---|---|---|
| Sequence-based reagent | hNG1-N41P-rev | This paper | PCR primers | GGGTTTCTGAGGATGGGGTCAGCATAGGTGA |
| Sequence-based reagent | hNG1-G79F80-for | This paper | PCR primers | GTTTATTTGAAATGGCCGCTGAAGAGGGAGAAAC |
| Sequence-based reagent | hNG1-G79F80-rev | This paper | PCR primers | GTTTCTCCCTCTTCAGCGGCCATTTCAAATAAAC |
| Sequence-based reagent | pIRES-EcoRI-huNGLY1-F | This paper | PCR primers | CTCAAGCTTCGAATTCTCAAGCATGGCGGCGGCG |
| Sequence-based reagent | v5His6-tga-SalI-pIRES-R | This paper | PCR primers | CCGCGGTACCGTCGACTCAATGGTGATGGTGGTGATGCGTAGAATCGAGACCGAG |
| Sequence-based reagent | NGLY1-N41P-F | This paper | PCR primers | ATTAGGGTTTCTGAGGATGGGGTCAGCATAGGTGAGCAGC |
| Sequence-based reagent | NGLY1-N41P-R | This paper | PCR primers | GCTGCTCACCTATGCTGACCCCATCCTCAGAAACCCTAAT |
| Sequence-based reagent | NGLY1-G79A_F80A-F | This paper | PCR primers | GATGAGATGTGTTTCTCCCTCTTCAGCGGCCATTTCAAATAAACATTCAA-CAGC |
| Sequence-based reagent | NGLY1-G79A_F80A-R | This paper | PCR primers | GCTGTTGAATGTTTATTTGAAATGGCCGCTGAAGAGGGAGAAACACATCTCATC |
| Sequence-based reagent | m-Psmb1-F | *Yang et al., 2018* | PCR primers | CCTTCAACGGAGGTACTGTATTG |
| Sequence-based reagent | m-Psmb1-R | *Yang et al., 2018* | PCR primers | GGGCTATCTCGGGTATGAATTG |
| Sequence-based reagent | m-Psmb4-F | *Yang et al., 2018* | PCR primers | CGAGTCAACGACAGCACTAT |
| Sequence-based reagent | m-Psmb4-R | *Yang et al., 2018* | PCR primers | ATCTCCCAACAGCTCTTCATC |
| Sequence-based reagent | m-Psma7-F | *Yang et al., 2018* | PCR primers | CGAGTCTGAAGCAGCGTTAT |
| Sequence-based reagent | m-Psma7-R | *Yang et al., 2018* | PCR primers | AGTCTGATAGAGTCTGGGAGTG |
| Peptide, recombinant protein | Recombinant Human BMP-4 Protein | R and D Systems | Cat# 314 BP | |
| Commercial assay or kit | In-Fusion HD Cloning | TaKaRa | Cat# 638920 | |
| Commercial assay or kit | Site-Directed Mutagenesis Kit | Agilent Technologies | Cat# 200523 | |

*Appendix 1—key resources table continued*

| Reagent type (species) or resource | Designation | Source or reference | Identifiers | Additional information |
|---|---|---|---|---|
| Commercial assay or kit | ENDEXTTechnology, Protein Research Kit (H) | Cell Free Science | CFS-PRK-H16 | |
| Commercial assay or kit | HA-tag IP/Co-IP kit | PIERCE | Cat# 26180 | |
| Chemical compound, drug | NMS-873 | Sigma-Aldrich | Cat# 5310880001 | |
| Chemical compound, drug | Bortezomib | Cayman Chemical | Cat# 10008822 | |
| Software, algorithm | GraphPad Prism | Prism | RRID:SCR_002798 | |

