## [Decision Letter]

Thank you for submitting your article "Regulation of BMP4/Dpp retrotranslocation and signaling by deglycosylation" for consideration by *eLife*. Your article has been reviewed by two peer reviewers, and the evaluation has been overseen by K VijayRaghavan as the Senior and Reviewing Editor. The reviewers have opted to remain anonymous.

The reviewers have discussed the reviews with one another and the Reviewing Editor has drafted this decision to help you prepare a revised submission.

Summary:

The manuscript by Galeone et al. describes a novel mechanism by which Dpp/BMP4 signaling is regulated. These findings are novel and important on several fronts. First, they provide a mechanistic explanation for the authors previous studies showing that NGLY1 is required for Dpp signaling in select tissues. Second, they demonstrate that NGLY1 is also required for BMP signaling in mammals, which might explain some of the phenotypic defects observed in NGLY1 deficient humans. Third, they demonstrate that, at least in some cases, NGLY1 is required for retrotranslocation of misfolded proteins out of the ER, in contrast to the accepted model in which cytoplasmic NGLY1 deglycosylates misfolded proteins after they have been removed from the ER. The manuscript is clearly written and the data are of high quality.

In a 2017 paper in *eLife*, Galeone et al. demonstrated that the loss of Pngl/NGLY1 in *Drosophila* causes midgut defects due to defective Dpp/BMP signaling. This was linked to the enzymatic activity of Pngl since catalytically-inactive Pngl was unable to rescue the BMP signaling. However, it remained unknown if Pngl was acting on Dpp directly or if this was an indirect effect mediated through other substrates of Pngl. In the current manuscript, the authors further explore this phenomenon and work out the mechanistic details. The authors provide compelling evidence to demonstrate that Dpp and its mammalian ortholog BMP4 are subject to deglycosylation by Pngl/NGLY1 without which BMP signaling is defective. Also, the authors show that this deglycosylation event is a pre-requisite for retrotranslocation of (presumably) misfolded BMP4 from the endoplasmic reticulum (ER). This retrotranslocation step is facilitated by the presence of ER-associated NGLY1. This is an interesting observation since NGLY1 is traditionally thought to act on substrates in the cytosol after their dislocation from the ER. Overall, this study makes a number of mechanistic advances both in the BMP4 realm as well as the ER-associated degradation (ERAD) area.

Essential revisions:

In order to better support the conclusions and increase the rigor of the study the following points need clarification

1) The authors state that accumulation of "misfolded" BMP4 in the ER results in ER stress and recruitment of NGLY1. Is there evidence to suggest that any BMP4 is actually misfolded?

2) The authors' assertion on ER stress in Figure 4 relies on a single marker GRP78. Additional ER stress markers will strengthen this point.

3) Figure 3C should directly demonstrate equal levels of BMP4-HA-Myc in the transfected samples, instead of relying on co-transfected GFP.

4) Figure 5 shows a correlation between NGLY1 mutants unable to VCP and their loss of recruitment to the ER. This can be further strengthened by demonstrating this by at least one other method, eg. Microsome isolation followed by immunoblots for VCP and different NGLY1 mutants.

5) Since one of the main points of the paper is that BMP4 is an ERAD substrate, demonstrating this aspect by chemical inhibition and/or genetic depletion of VCP would strengthen the case.

6) There is a heavy reliance on tagged overexpressed BMP4. This is understandable since BMP4 may not be expressed sufficiently in all cell types. However, in order to rule out overexpression artifacts, is there a way to demonstrate the key aspects of this paper by examining endogenous BMP4?

---

## [Author Response]

Essential revisions:In order to better support the conclusions and increase the rigor of the study the following points need clarification1) The authors state that accumulation of "misfolded" BMP4 in the ER results in ER stress and recruitment of NGLY1. Is there evidence to suggest that any BMP4 is actually misfolded?

We thank the reviewer for raising this important point. In the original version, we had relied on the increased level of GRP78/BiP as evidence for BMP4 misfolding in the ER. Moreover, as shown in the original Figure 6D, the deglycosylated BMP4 increased upon proteasomal inhibition, suggesting that deglycosylated BMP4 is retrotranslocated to the cytosol for degradation. However, we agree that this crucial point of the manuscript needed to be addressed with more direct evidence. While addressing point #2 (please see below), we found that transfection of BMP4-HA-Myc into control (*Ngly1^+/+^*) cells results in a moderate increase in the level of OS9, an ER lectin which selectively binds misfolded glycoproteins in the ER and facilitates their transport to the retrotranslocation machinery. OS9 levels were further increased in *Ngly1^–/–^* cells transfected with BMP4-HA-Myc (please see revised Figure 4B). Given the specific binding of OS9 to misfolded glycoproteins in the ER, these results provided an opportunity for us to directly address the issue of BMP4 misfolding by asking whether BMP4 binds OS9 in *Ngly1^–/–^* cells. Our new co-IP experiments showed that in *Ngly1^–/–^* MEFs, BMP4HA-Myc molecules are in a complex with OS9. Cotransfection of *Ngly1^–/–^* MEFs with wild-type human NGLY1 and BMP4-HA-Myc reduced the OS9 levels in these cells, proving that OS9 accumulation was due to the loss of NGLY1. Importantly, when VCP-binding mutant versions of NGLY1 were cotransfected with BMP4-HA-Myc into *Ngly1^–/–^* MEFs, the level of OS9 in these cells and its interaction with BMP4 remained high. These data, which are shown in revised Figure 7A, provide strong evidence that misfolded BMP4 accumulates in the ER upon loss of NGLY1, or upon a failure in NGLY1 recruitment to the ER.

In addition to these experiments, the new data that we are providing in response to points #2 and #5 provide further evidence for misfolding of BMP4 in the ER. Specifically, as shown in revised Figure 4, BMP4-HA-Myc transfection leads to an increase in the level of the phosphorylated IRE1α both in *Ngly1^+/+^* and *Ngly1^–/–^* cells, an observation compatible with BMP4 misfolding. Moreover, as shown in new Figure 7B, treating *Ngly1^+/+^* cells with a VCP inhibitor (NMS-873) resulted in the accumulation of glycosylated BMP4-HA-Myc, a reduction in deglycosylated BMP4-HA-Myc, and induction of GRP78/BiP. These observations indicate that BMP4 is an ERAD substrate in *Ngly1^+/+^* cells that depends on the function of VCP to gain access to the deglycosylation activity of NGLY1, further supporting the notion that some BMP4 molecules are misfolded in the ER.

2) The authors' assertion on ER stress in Figure 4 relies on a single marker GRP78. Additional ER stress markers will strengthen this point.

As shown in the revised Figure 4 and discussed above, *Ngly1^–/–^* MEFs also show an increase in the level of the phosphorylated IRE1α, which is an indication of unfolded protein response activation, and an increase in the level of OS9, which is an ER lectin specifically binding misfolded glycoproteins. Both of these are also induced when BMP4 is overexpressed in *Ngly1^+/+^* MEFs and further increased when BMP4 is overexpressed in *Ngly1^–/–^* MEFs. These observations provide additional evidence for ER stress in this model.

3) Figure 3C should directly demonstrate equal levels of BMP4-HA-Myc in the transfected samples, instead of relying on co-transfected GFP.

We agree that the level of BMP4-HA-Myc would have been the ideal control for this experiment. However, as shown in Figure 4D, loss of NGLY1 results in accumulation of a glycosylated form of BMP4 despite comparable transfection (as evidenced by GFP) and comparable loading (as evidenced by Tubulin). Therefore, we had to rely on these two markers (GFP and Tubulin) to normalize the data shown in Figure 3C. We note that the first two lanes of Figure 4E provide another example of this experiment and show that the level of pSMAD1/5 is severely decreased in *Ngly1^–/–^* MEFs despite the presence of high levels of glycosylated BMP4-HA-Myc in these cells.

4) Figure 5 shows a correlation between NGLY1 mutants unable to VCP and their loss of recruitment to the ER. This can be further strengthened by demonstrating this by at least one other method, eg. Microsome isolation followed by immunoblots for VCP and different NGLY1 mutants.

As suggested by the reviewers, we have obtained a microsome isolation kit and an antibody against VCP to address this point. However, we need additional time to set up this rather challenging experiment and to repeat the experiment a number of times to make sure our observations are reproducible, as we have done for all other experiments. We plan to perform this experiment in the coming weeks, unless additional limits are placed on our laboratory access due to COVID-19. If these experiments are successful, we will deposit the data on BioRxiv, or combine it with other data to submit it as another Research Advance to *eLife*, as described in a recent Commentary by *eLife* Editors and recommended to us by Dr. VijayRaghavan.

5) Since one of the main points of the paper is that BMP4 is an ERAD substrate, demonstrating this aspect by chemical inhibition and/or genetic depletion of VCP would strengthen the case.

To address this point, we transfected *Ngly1^+/+^* MEFs with BMP4-HA-Myc and treated them with the VCP inhibitor NMS-873, the proteasome inhibitor Bortezomib (BTZ), or both. As shown in the revised Figure 7B, NMS-873 treatment led to the accumulation of glycosylated BMP4-HA-Myc in MEFs, accompanied by a reduction in the deglycosylated BMP4-HA-Myc and an increase in the level of the ER chaperone GRP78/BiP. BTZ alone resulted in the accumulation of deglycosylated BMP4-HA-Myc, in agreement with data in the original Figure 6D (shown in the revised version as well). Combination of NMS-873 and BTZ resulted in the accumulation of deglycosylated BMP4-HA-Myc and an enhanced accumulation of glycosylated BMP4-HA-Myc compared to cells treated with NMS-873 alone. Together, these data provide strong evidence that BMP4 is an ERAD substrate.

6) There is a heavy reliance on tagged overexpressed BMP4. This is understandable since BMP4 may not be expressed sufficiently in all cell types. However, in order to rule out overexpression artifacts, is there a way to demonstrate the key aspects of this paper by examining endogenous BMP4?

Our results on endogenous fly Dpp and the dramatic reduction in pSMAD1/5 staining of the 4th ventricle choroid plexus (where BMP4 is known to play a key role) in *Ngly1^–/–^* mouse brains provided evidence for the role of NGLY1 on endogenous Dpp/BMP4. However, as the reviewers mentioned, our conclusions about the role of NGLY1 on BMP4 signaling from MEFs were based on overexpression. To address this concern, we reviewed the literature and learned that the mouse preadipocyte cell line 3T3L1 has been shown to express significant levels of *Bmp4* (but not *Bmp2* and *Bmp7*). Furthermore, BMP4 knock-down in these cells resulted in a severe reduction of pSMAD1/5 and affected the adipocyte fate in these cells (Suenaga et al., 2010; Suenaga et al., 2013). Accordingly, we decided to use these cells to ask whether endogenous BMP4 signaling in a mammalian cell line requires NGLY1. To this end, we used a well-established pharmacological inhibitor of NGLY1 called Z-VADfmk. As shown in the new Figure 3—figure supplement 1, both low (20 μM) and high (50 μM) doses of Z-VAD-fmk resulted in a statistically significant reduction of pSMAD1/5 levels in these cells. High levels of Z-VAD-fmk are known to inhibit caspases as well. However, a high dose of an independent broad spectrum caspase inhibitor that does not affect NGLY1 (Q-VD-OPh) did not affect pSMAD1/5 levels in 3T3-L1 cells. These observations provide strong evidence that the effects of loss of NGLY1 on BMP4 signaling is not limited to overexpression contexts.

As mentioned above, endogenous BMP4 was previously shown to be the major BMP ligand in 3T3-L1 cells by Suenaga and colleagues. However, we would have liked to directly show the effect of NGLY1 inhibition on the BMP4 protein in these cells. Unfortunately, Western blotting with a commercially available BMP4 antibody generated multiple bands at various sizes from 3T3-L1 cell extracts, which prevented us from pursuing this avenue. To follow the fate of endogenous BMP4 in 3T3-L1 upon NGLY1 inhibition, one would ideally need to use CRISPR to tag the endogenous BMP4 similar to our BMP4-HA-Myc construct. However, we believe that this would be beyond the scope of the current work.